# Reactive oxygen species activate the *Drosophila* TNF receptor Wengen for damage-induced regeneration

José Esteban-Collado[1,2], Mar Fernández-Mañas [1,5], Manuel Fernández-Moreno [1,3,4], Ignacio Maeso [1,3], Montserrat Corominas [1,2] & Florenci Serras [1,2✉]

## Abstract

Tumor necrosis factor receptors (TNFRs) control pleiotropic pro-inflammatory functions that range from apoptosis to cell survival. The ability to trigger a particular function will depend on the upstream cues, association with regulatory complexes, and downstream pathways. In *Drosophila melanogaster*, two TNFRs have been identified, Wengen (Wgn) and Grindelwald (Grnd). Although several reports associate these receptors with JNK-dependent apoptosis, it has recently been found that Wgn activates a variety of other functions. We demonstrate that Wgn is required for survival by protecting cells from apoptosis. This is mediated by dTRAF1 and results in the activation of p38 MAP kinase. Remarkably, Wgn is required for apoptosis-induced regeneration and is activated by the reactive oxygen species (ROS) produced following apoptosis. This ROS activation is exclusive for Wgn, but not for Grnd, and can occur after knocking down Eiger/TNFα. The extracellular cysteine-rich domain of Grnd is much more divergent than that of Wgn, which is more similar to TNFRs from other animals, including humans. Our results show a novel TNFR function that responds to stressors by ensuring p38-dependent regeneration.

**Keywords** Apoptosis; Cell Signaling; Cytokines; Oxidative Stress; Regeneration
**Subject Categories** Autophagy & Cell Death; Signal Transduction

See also: DS Andersen & J Colombani

## Introduction

The cytokine tumor necrosis factor alpha (TNFα) is rapidly released after trauma, injury, or infection and acts as a central mediator in the inflammatory response. It signals through the TNF receptors (TNFRs) and its function is currently understood to be pleiotropic, playing a role in apoptosis, cell survival, and cell proliferation, the outcome of which will be determined by different TNFRs and complex signaling networks (Gough and Myles, 2020).

Oxidative stress, generated by the production of reactive oxygen species (ROS), is recognized as an underlying cause of a variety of inflammatory diseases (Conner and Grisham, 1996; Droge, 2002; Finkel, 2011). ROS produce deleterious effects on cells because of their ability to oxidize cell structures, such as DNA (Van Houten et al, 2018), membrane lipids and proteins (Colquhoun, 2010; Moriarty-Craige and Jones, 2004). However, it is increasingly well-accepted that ROS-induced post-translational modifications of proteins may also be of physiological relevance in cell signaling. Among these is the oxidation of cysteine residues in receptor proteins, which results in their activation (Lipton et al, 2002; Truong and Carroll, 2012). Likewise, oxidative stress can modify the reduced thiol groups of the TNFR1 extracellular cysteine-rich domain (CRD), which is the hallmark domain shared by the TNFR superfamily (Dominici et al, 2004). This raises the intriguing possibility that oxidative stress might modulate TNFR signaling. Indeed, oxidative stress promotes the self-association of the subunits of mammalian TNFR1 and 2, which results in ligand-independent signaling as well as enhanced ligand-dependent TNF signaling (Ozsoy et al, 2008).

Tolerable levels of ROS can propagate as paracrine signals and modulate the intracellular machinery that will reconstruct the damaged tissues during regeneration (Brock et al, 2017; Diwanji and Bergmann, 2018; Farrell et al, 2022; Fogarty et al, 2016; Khan et al, 2017; Patel et al, 2019; Pérez et al, 2017; Santabárbara-Ruiz et al, 2015; Serras, 2022; Weavers et al, 2019). In the presence of ROS, thioredoxin dissociates from the MAPKKK Apoptosis signal-regulating kinase 1 (Ask1), following which Ask1 oligomerizes, autophosphorylates, recruits its partners and forms an active kinase complex (Bunkoczi et al, 2007; Matsuzawa, 2016; Obsil and Obsilova, 2017; Takeda et al, 2008). An active Ask1 catalytic domain triggers the phosphorylation of the MAPKK that in turn phosphorylate JNK and p38 to induce apoptosis (Bunkoczi et al, 2007; Ichijo et al, 1997; Tobiume et al, 2001). But Ask1 has other functions beyond apoptosis. In *Drosophila*, Ask1 operates in the gut and imaginal disc epithelia as a survival signal in tissue regeneration by triggering a non-apoptotic function of p38 (Patel et al, 2019; Santabárbara-Ruiz et al, 2019). This is achieved by a ROS-dependent, Akt-induced phosphorylation of Ser83 in Ask1

[1]Department of Genetics, Microbiology and Statistics, Faculty of Biology, University of Barcelona, Barcelona, Spain. [2]Institut de Biomedicina de la Universitat de Barcelona (IBUB), Barcelona, Spain. [3]Institute for Biodiversity Research (IRBio), Barcelona, Spain. [4]Centre for Genomic Regulation (CRG), Barcelona, Spain. [5]Present address: Institut d'Investigacions Biomèdiques August Pi i Sunyer (IDIBAPS), Barcelona, Spain.✉E-mail: fserras@ub.edu

(Ser174 in humans) that is essential to redirect the Ask1 signaling pathway towards p38 and thus drive regeneration (Esteban-Collado et al, 2021).

The kinase activity of Ask1 is also stimulated by TNFα via members of the TNF Receptor-Associated Factor (TRAF) family of adapter proteins (Ichijo et al, 1997; Nishitoh et al, 1998). Upon the binding of TNFα to TNFRs, a subsequent interaction with TRAFs occurs, facilitating the transduction of the signal to Ask1 and thereby modulating its activity (Hoeflich et al, 1999; Nishitoh et al, 1998; Obsil and Obsilova, 2017; Shiizaki et al, 2013). How this TNF/TNFR/TRAF axis drives apoptosis or survival is poorly understood. In contrast with the large families of both TNF and TNFRs in mammals, *Drosophila* has only one TNFα ortholog, *eiger* (*egr*), and two TNFRs, *wengen* (*wgn*) and *grindelwald* (*grnd*). Grnd mediates the pro-apoptotic function of Egr/TNFα, and its overexpression activates JNK-dependent apoptosis (Andersen et al, 2015; Palmerini et al, 2021). Wengen was first discovered as a TNFR that is able to transduce signals from Egr/TNFα, bind to TRAFs and trigger JNK-dependent apoptosis (Geuking et al, 2005; Kanda et al, 2002; Kauppila et al, 2003). However, it has been shown that while the knockdown of *grnd* blocks apoptosis, that of *wgn* does not (Andersen et al, 2015), suggesting that *wgn* and *grnd* are not redundant. Moreover, both TNFRs are transmembrane proteins that form hexamers for ligand-joining, but Wgn binds to Egr/TNFα at an affinity that is three orders of magnitude lower than Grnd (Palmerini et al, 2021). Recent work has demonstrated that in the gut, Wgn suppresses dTRAF3-mediated lipolysis independently of its ligand and maintains tissue homeostasis, while Wgn-dTRAF2-mediated immune suppression occurs in an Egr-dependent manner (Loudhaief et al, 2023). In addition, the ligand-independent function of Wgn has been recently demonstrated to associate with unrelated factors such as FGFR to regulate vesicle trafficking during tracheal development (Letizia et al, 2023). Wgn is also expressed in photoreceptor progenitors and binds to moesin for axonal pathfinding in a ligand-independent manner (Ruan et al, 2013). Moreover, *Drosophila* TNFRs show a different subcellular localization. Grnd is mainly found on the apical side of epithelial cells and becomes internalized upon binding to Egr, whereas Wgn is mainly found in intracellular vesicles (Andersen et al, 2015; Letizia et al, 2023; Loudhaief et al, 2023). So far, the evolutionary origin of the two *Drosophila* TNFRs and their relationship to TNFRs from other species have not been investigated in detail, limiting our understanding of how the contrasting functions and molecular behavior of these two receptors may have originated.

Therefore, we used the wing imaginal disc epithelium to explore whether these *Drosophila* TNFRs are involved in survival or apoptosis, whether they respond to oxidative stress and, ultimately, whether they are required for apoptosis-induced regeneration. Here we show that Wgn/dTRAF1 is required for cell survival, in contrast to the apoptotic role of Grnd/dTRAF2. Evolutionary analyses of TNFRs showed an independent and ancient origin of *grnd* and *wgn*, reinforcing the idea of a subfunctionalization of these genes and a higher degree of similarity between the CRDs of Wgn and the TNFRs of humans and other deuterostomes. Indeed, Wgn, but not Grnd, is required for regeneration and for the activation of p38 in the damage response. Interestingly, the activation of Wgn is sensitive to the ROS produced by damaged cells and is not affected by knocking down Egr/TNFα.

# Results

## Wgn is required for survival by protecting cells from apoptosis

We first studied the involvement of the TNFRs Grnd and Wgn in apoptosis and survival. The two *Drosophila* TNFRs are normally expressed in the wing imaginal discs (Palmerini et al, 2021), whereas Eiger/TNFα (*egr*) is not. Ectopic expression of *egr* in *Drosophila* imaginal discs results in JNK-dependent apoptosis (Brodsky et al, 2004; Igaki et al, 2002; Moreno et al, 2002b). We ectopically expressed *egr* in the wing disc, using the *Gal4/UAS/Gal80^{TS}* transactivation system, and simultaneously knocked down the TNFRs using appropriate RNAi strains. We used the *hedgehog-Gal4* strain (hereafter *hh>*) to activate the transcription of *egr* in the posterior compartment using the *UAS-egr^{weak}* strain (hereafter *egr^{weak}*), a transgene that causes mild/moderate *egr* overexpression (Moreno et al, 2002a).

The expression of *egr^{weak}* resulted in low levels of apoptotic cells in the posterior compartment of the disc (Figs. 1A,I and EV1A). However, *egr^{weak}* expression resulted in abolition of apoptosis when *grnd* was downregulated (Figs. 1B,I and EV1F), which coincides with previous findings that Grnd promotes JNK-dependent apoptosis (Andersen et al, 2015; Palmerini et al, 2021). In contrast, *egr^{weak}* expression resulted in a strong enhancement of apoptosis when *wgn* was downregulated (Fig. 1E,I). This observation was corroborated with an independent *wgn* RNAi strain targeting a different region in the coding sequence (Fig. EV1D,E). Remarkably, expression of any of the two *wgn* RNAi strains alone under the *hh>* driver produced only few scattered caspase positive cells (Fig. EV1D,E). Therefore, the massive *wgn*-related apoptosis occurs only in stressed conditions generated by *egr* expression, suggesting that *wgn* is recruited for a stress-dependent context.

Next, we knocked down the adapter protein *dTRAF2* and the MAPKKK *Tak1* and found reduced or similar levels of apoptosis compared to *egr^{weak}* alone, respectively (Figs. 1C,D,I and EV1G,H), confirming the role of those genes in mediating the pro-apoptotic function of TNF signaling. In contrast, knocking down *dTRAF1* and *Ask1* resulted in increased apoptosis in comparison to *egr^{weak}* alone (Figs. 1F,G,I and EV1I,J). Note that when *dTRAF2* or *grnd* is downregulated in *egr^{weak}* cells, the cell death area ratio is slightly lower than *egr^{weak}* alone (Fig. 1I), confirming that dTRAF2 and Grnd contribute to apoptosis in *egr^{weak}* cells. All these observations were corroborated with independent RNAi strains from different sources (Figs. 1I and EV1).

To check for epistasis between *grnd* and *wgn*, we activated *hh> egr^{weak}* and knocked down both TNFRs. We found high levels of cell death compared to *wgn* RNAi alone (Fig. 1H,I), which suggests that *wgn* downregulation is dominant over *grnd*. This is surprising, as it is generally assumed that Egr interacts with Grnd to induce apoptosis via JNK, which in turn activates the pro-apoptotic gene *hid* (Andersen et al, 2015; Diwanji and Bergmann, 2020; Fogarty et al, 2016; Igaki et al, 2002; Moreno et al, 2002b; Sanchez et al, 2019; Shlevkov and Morata, 2012). Interestingly, Egr is necessary for the stabilization of the pro-apoptotic gene *hid* and can regulate HID-induced apoptosis independently of JNK (Shklover et al, 2015). However, we cannot rule out the presence of residual Grnd/JNK signal, which may be enough to contribute to the apoptosis in the *grnd* and *wgn* double knockdown.

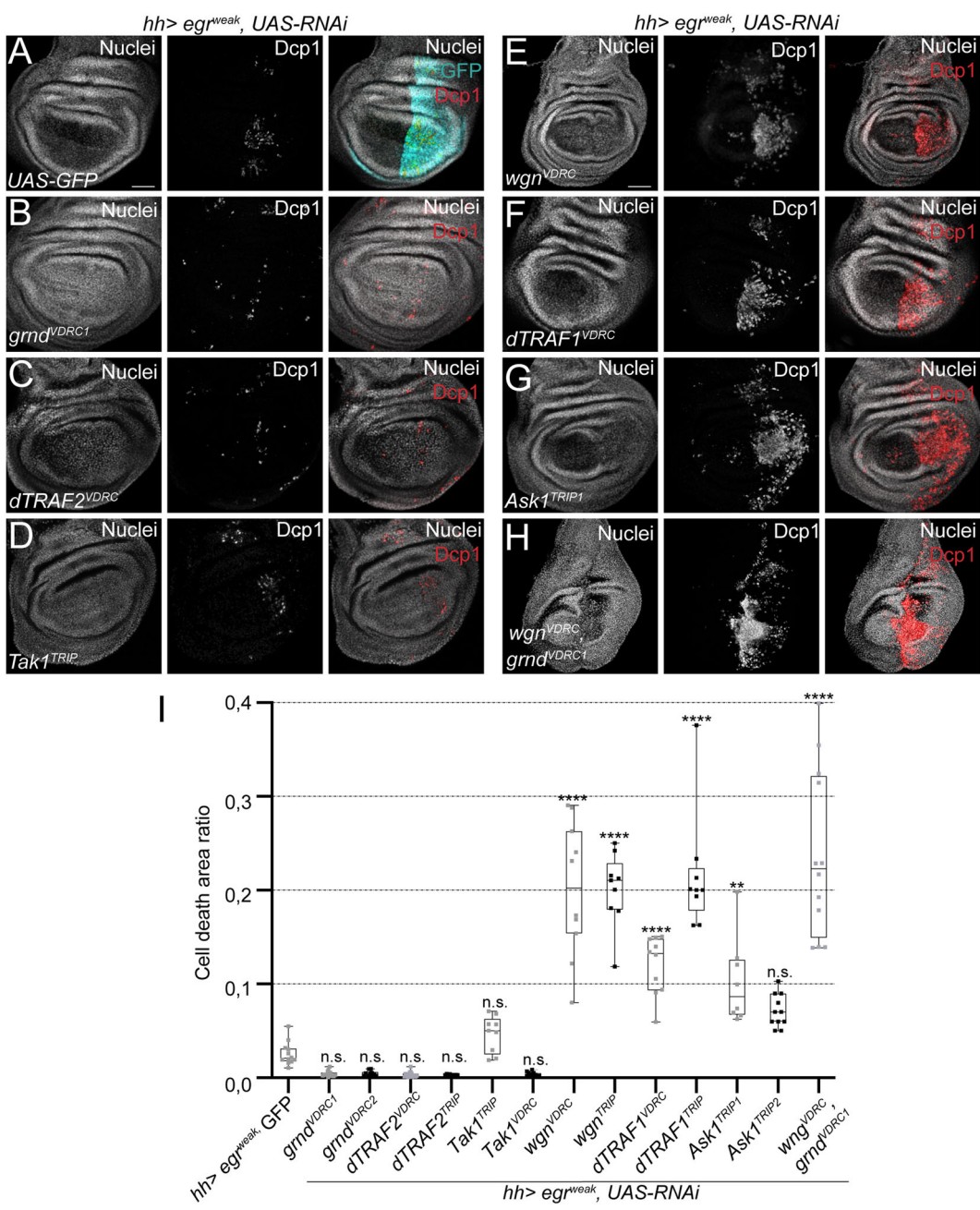

**Figure 1.  Induction of apoptosis in Egr/TNFα cells after downregulation of different TNFR pathway genes.**

(A–H) Dcp1-positive cells (red) after ectopic co-expression of *egr*^weak^ and (A) GFP-expressing control (*n* = 12), (B) *grnd* RNAi (*n* = 15), (C) *dTRAF2* RNAi (*n* = 18), (D) *Tak1* RNAi (*n* = 9), (E) *wgn* RNAi (*n* = 18), (F) *dTRAF1* RNAi (*n* = 10), (G) *Ask1* RNAi (*n* = 8) and (H) double RNAi of *wgn* and *grnd* (*n* = 12). All transgenes were ectopically expressed in the posterior compartment (*hh*>; cyan in A). TP3 was used to stain nuclei. Scale bar: 50 μm. (I) Cell death area ratio calculated in base 1 for the genotypes indicated. Box plots show maximum–minimum range (whiskers), upper and lower quartiles (open rectangles), and median value (horizontal black line), each dot representing the cell death area ratio from a different imaginal disc. Gray dots: a disc of each genotype is shown in (A–H); Black dots: a disc of each genotype in EV1. One-way ANOVA test was used for multiple comparisons between all groups: n.s. = not significant, **$p$ = 0.0002, ****$p$ < 0.0001. Source data are available online for this figure.

These results show that *wgn* is required in cell survival, likely through the Wgn/DTRAF1/Ask1 signaling cassette. In addition, as *grnd* has been demonstrated to be involved in JNK-dependent apoptosis (Andersen et al, 2015), our observations also indicate that the functions of *wgn* and *grnd* are essentially different.

## Wgn is required for the activation of the p38 MAP kinase

Although Ask1 can activate JNK and p38 (Tobiume et al, 2001), stimulation of p38 and inhibition of the JNK-pro-apoptotic function is necessary for a regenerative response to damage (Patel

et al, 2019; Santabárbara-Ruiz et al, 2019). Therefore, we hypothesized that the survival role driven by Wgn is likely accomplished by p38 rather than JNK.

To explore this hypothesis, we tested if the apoptosis induced by knocking down *wgn* can be abolished by ectopic stimulation of JNK or p38. We used *hh>egr^weak, wgn RNAi*, which exacerbated apoptosis, and expressed the MAPKK (*UAS-hep^WT* or *UAS-lic^WT*) upstream of either JNK or p38. An ectopic activation of the JNK pathway by *UAS-hep^WT* resulted in increased levels of apoptosis in comparison to the control (*wgn RNAi* alone) (Fig. 2A,C,D). In contrast, an ectopic activation of p38 by *lic^WT* resulted in a strong decrease in apoptosis (Fig. 2A,B,D), suggesting that activated p38 functions downstream of Wgn.

Additionally, p38 RNAi exacerbates apoptosis in *egr^weak* cells (Fig. EV2A). While *lic^WT* alone results in few scattered apoptotic cells (Esteban-Collado et al, 2021), expression of *lic^WT* in *egr^weak* cells induced extensive apoptosis (Fig. EV2B), possibly due to the excess of active p38 generated by the addition of phospho-p38 in response to *egr^weak* and the activity of *lic^WT*. These observations support that the survival role of p38 must be finely controlled, as an excess or shortage of p38 activity is detrimental to the tissue (Esteban-Collado et al, 2021).

## Wgn is activated by ROS

Next, we investigated whether Wgn responds to the stress generated by the expression of *egr*. To this end, we first examined the localization of Wgn and Grnd after *egr^weak* expression. It has been reported that the majority of Wgn is localized in the cytoplasm in many organs, likely in intracellular vesicles rather than at the plasma membrane (Letizia et al, 2023; Loudhaief et al, 2023; Palmerini et al, 2021). By contrast, Grnd is localized in the plasma membrane, making it more accessible to Egr/TNFα (Palmerini et al, 2021). We confirmed these localization patterns for Wgn and Grnd in control imaginal discs (Fig. 3A,B).

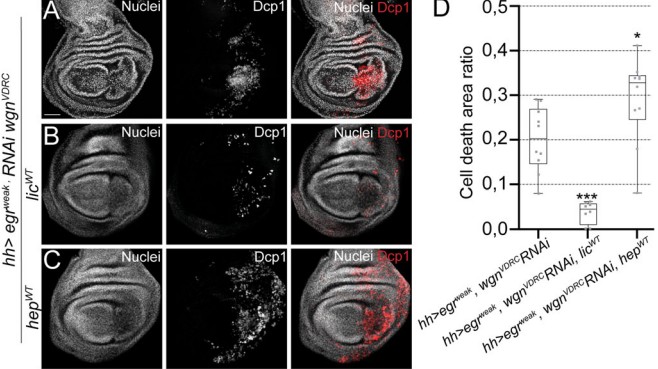

**Figure 2. The p38 MAPKK *lic* rescues the *wgn* mutant phenotype.**

(A–C) Dcp1-positive cells after *egr^weak* overexpression and *wgn RNAi* in (A) *wgn RNAi* (n = 10), (B) *wgn RNAi* and *lic^WT* (n = 8), (C) *wgn RNAi* and *hep^WT* (n = 10). (D) Cell death area ratio calculated in base 1 for the genotypes indicated. Box plots show maximum–minimum range (whiskers), upper and lower quartiles (open rectangles), and median value (horizontal black line). *p = 0.0314, ***p = 0.0002. TP3 was used to stain nuclei. Scale bar: 50 μm. Source data are available online for this figure.

After ectopic *egr^weak* expression in the posterior compartment, Wgn particles were found accumulated in the cytoplasm of the anterior compartment, particularly in anterior cells close to the border with the posterior compartment, and they were absent in the *egr^weak* posterior compartment (Fig. 3C). Apoptotic cells in the latter compartment (*egr^weak*) are characterized by pyknotic nuclei and are positive for Dcp1. These cells can be found along the apical-basal axis, although they eventually tend to concentrate on the basal side of the epithelium (Fig. EV3A,B). An accumulation of Wgn was observed in healthy anterior cells adjacent to both apical and basal *egr^weak* cells (Figs. 3C and EV3A,B). By contrast, Grnd was maintained on the apical membrane of the anterior compartment and was found internalized in the *egr^weak* posterior compartment (Fig. 3D), which aligns with previous observations that Grnd is translocated from the membrane after binding to Egr/TNFα (Andersen et al, 2015).

It is known that apoptosis generates oxidative stress due to the production of ROS of mitochondrial origin that can propagate to recruit neighboring cells for damage repair (Serras, 2022). Furthermore, the oxidation of the TNFR CRD by ROS is a physiological mechanism able to transduce the signal (Ozsoy et al, 2008). Thus, to investigate the molecular mechanism underlying the accumulation of Wgn in neighboring cells after *egr* expression, we first checked if ROS were produced after *hh>egr^weak*. We used in vivo imaging with the cell-permeant fluorogenic probe MitoSOX, which is non-fluorescent in the reduced state and exhibits bright fluorescence upon oxidation by mitochondrial superoxide. MitoSOX-positive cells were found in the *egr^weak* compartment and they co-localized with cells positive for TO-PRO-3, a nucleic acid stain that is very sensitive to dead and dying cells (Fig. 3E). MitoSOX co-localization with TO-PRO-3 cells was also detected in *egr^weak* cells after knocking down *wgn*, indicating that inhibition of *wgn* does not block ROS production (Fig. EV3C). To discern whether ROS production responds to *egr^weak* expression or to apoptosis, we co-expressed *egr^weak* with the baculovirus protein p35, which blocks the effector caspases (Hay and de Belleroche, 1994). We found that neither MitoSOX nor TO-PRO-3 were detected when apoptosis was blocked in *egr^weak*-expressing cells (Fig. 3F). This suggests that the accumulation of ROS is caused by apoptosis rather than the expression of *egr*.

Next, we enzymatically decreased ROS production using ectopic expression of the ROS scavengers *Superoxide dismutase 1* and *Catalase* (*UAS-Sod1:UAS-Cat*). This resulted in a reduction of ROS (Fig. 3G) and the accumulation of Wgn particles near the ablated area compared to *egr^weak* alone (Fig. 3H). By contrast, Grnd localization was not altered following ROS depletion (Fig. 3I).

It is worth noting that *egr^weak* cells induce the phosphorylation of p38 in neighboring cells (Fig. EV3D), as occurs in cells that do not express the pro-apoptotic gene *reaper* (*rpr*) (Santabárbara-Ruiz et al, 2015). ROS depletion and p35 expression reduce phospho-p38 levels (Fig. EV3E,F). This suggests that p38 activation depends on ROS generated by apoptotic *egr^weak* cells.

We also induced apoptosis in an Egr-independent manner to monitor the localization of both TNFRs. With this aim, we expressed *rpr* using the *sal^E/Pv-Gal4* driver, whose expression is restricted to the central part of the wing imaginal disc (henceforth *sal^E/Pv >*). As a result, Wgn accumulated in the cells surrounding the apoptotic zone (Figs. 4A,G and EV4A). Contrastingly, apical Grnd localization in the tissue surrounding the

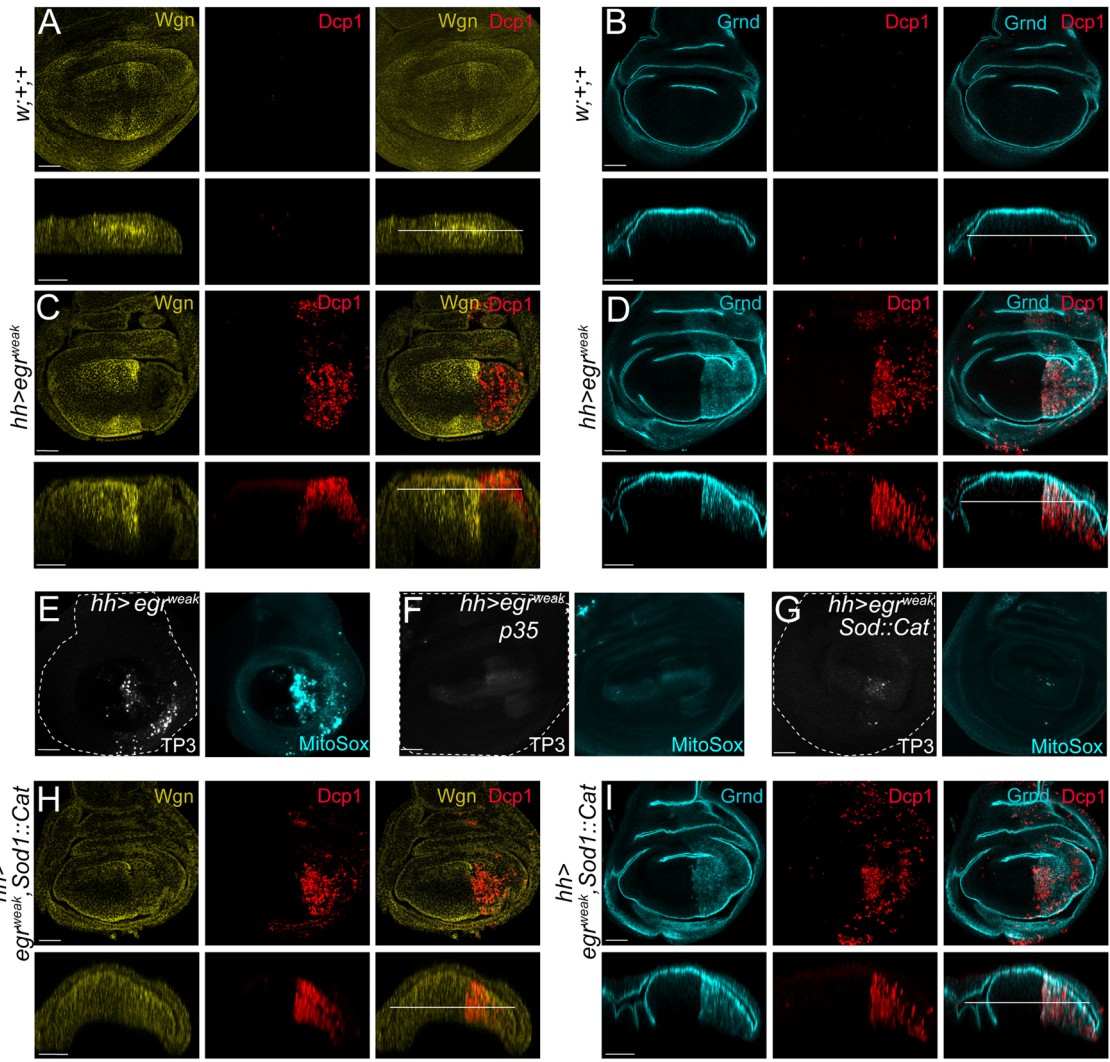

**Figure 3. Wgn accumulation after ectopic *egr/TNFα* expression is abrogated after enzymatic depletion of ROS.**

All transgenes were ectopically expressed in the posterior compartment (*hh>*). The wing pouch of each genotype (above) is complemented with an orthogonal apico-basal section (below); white lines indicate the level of the z-axis of the above images. (**A**) Anti-Wgn and Dcp1 staining of control wing discs (*n* = 11). (**B**) Anti-Grnd and Dcp1 staining of control wing discs (*n* = 4). (**C**) Anti-Wgn and Dcp1 staining of *egr^weak^* (*n* = 10). (**D**) Anti-Grnd and Dcp1 staining of *egr^weak^* (*n* = 7). (**E–G**) Discs stained ex vivo after *egr^weak^* activation. Left: cell death (TP3). Right: ROS of mitochondrial origin (MitoSOX) in (**E**) *wgn RNAi*, (**F**) *wgn RNAi* and ectopic expression of p35, (**G**) *wgn RNAi* and ectopic expression of *Sod1:Cat*; dotted lines indicate the contour of the imaginal discs. (**H**) Anti-Wgn and Dcp1 staining of *egr^weak^* and *Sod1:Cat* (*n* = 8). (**I**) Anti-Grnd and Dcp1 staining of *egr^weak^* and *Sod1:Cat* (*n* = 8). Scale bars: 50 μm. Source data are available online for this figure.

apoptotic zone did not vary (Fig. 4B,H). Wgn and Grnd were also detected in cellular debris in the apoptotic *rpr* expression zone. Moreover, we also reduced ROS production in the *rpr*-ablated region (*sal^E/Pv^>rpr, Sod1:Cat*) and found a decrease in Wgn, but not Grnd, levels in the cells surrounding the ablated zone (Figs. 4C,D,G,H and EV4B).

In addition, we found that *egr/TNFα* is autonomously expressed after inducing apoptosis by *rpr* (Fig. EV4D,E). To test if the effects on Wgn localization were due to ROS or to the expression of *egr*, we used RNAi to knock down *egr* in the apoptotic cells and found that Wgn accumulation was not altered by knocking down Egr/TNFα (Figs. 4E–H and EV4C).

Together, these observations suggest that the Wgn response to apoptotic ROS production occurs independently of Egr/TNFα.

## Wgn, but not Grnd, is necessary for p38-dependent regeneration

Regeneration in the gut and in imaginal discs depends on p38 in a ROS-dependent manner (Patel et al, 2019; Santabárbara-Ruiz et al, 2019). Therefore, we wondered whether Wgn is necessary for p38-dependent regeneration. We used a double transactivation system to simultaneously induce apoptosis in one domain of the wing disc to stimulate regeneration and to knock down either *wgn* or *grnd* in adjacent regenerating cells (Fig. 5A). We first confirmed that the mutants for *wgn* and *grnd* used in this work do not affect normal growth and patterns (Fig. EV5A). However, knocking down *wgn* after inducing apoptosis resulted in anomalous wings, a characteristic of incomplete regeneration (Fig. 5B). Most of these wings

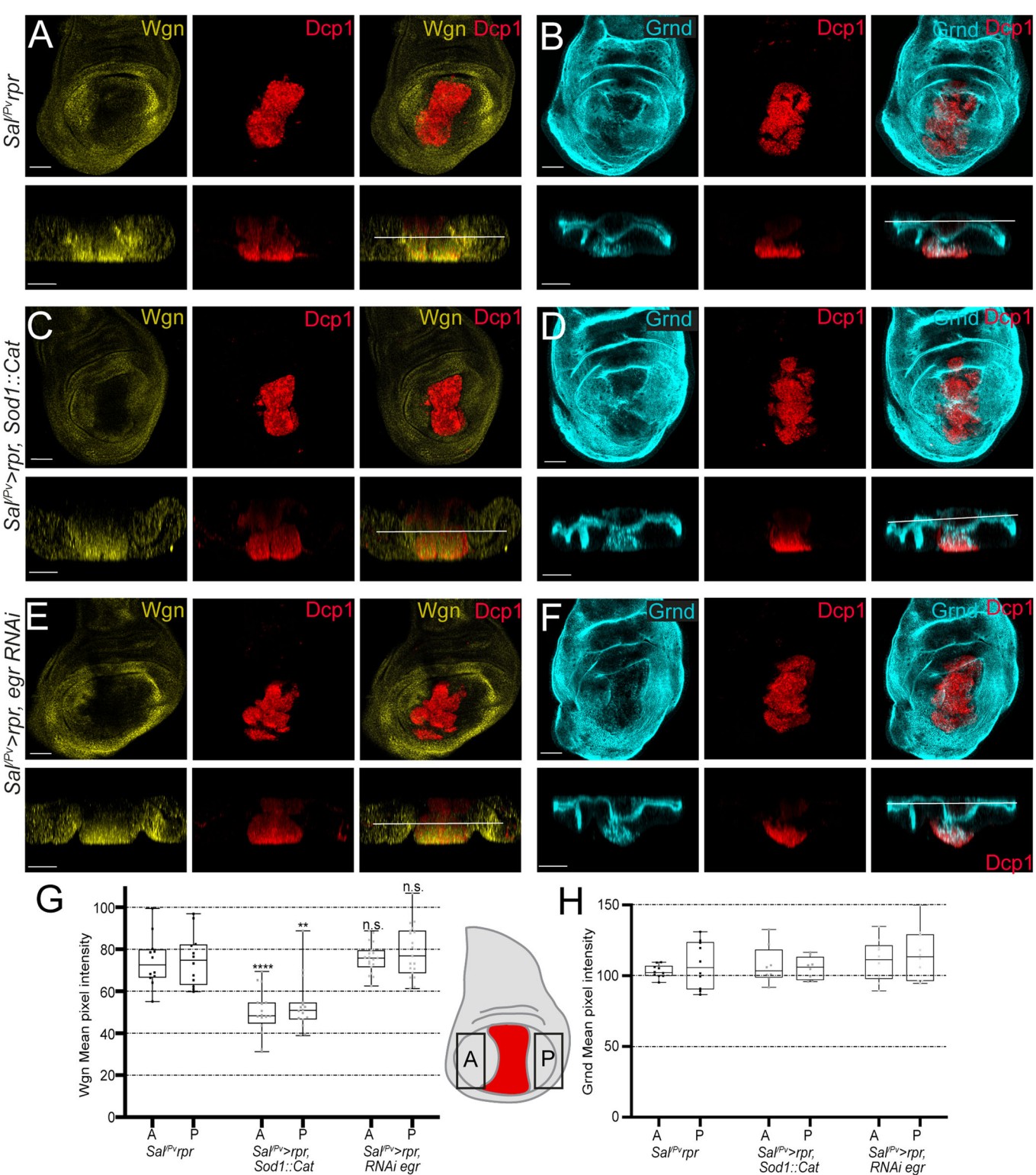

showed a reduced size and defective patterns of veins and interveins. By contrast, wings carrying a knock down of *grnd* did not show a reduced size or defects in wing patterning (Fig. 5B), suggesting that regeneration of the apoptotic zone was completed even in the absence of *grnd*. To confirm this observation, we induced cell death (*sal*^E/Pv^>*rpr*) in an independent *grnd* mutant,

*grnd*^minos^, in homozygosis and heterozygosis and found that wings regenerated normally. This result demonstrates that *wgn*, but not *grnd*, is necessary for the regenerative response after apoptosis.

Next, to test if *wgn* is required for p38-driven regeneration, we analyzed if phosphorylated p38 is affected by *wgn* downregulation after inducing apoptosis. We used the double transactivation

**Figure 4. Wgn accumulation after ectopic activation of the pro-apoptotic gene *rpr* is abrogated by enzymatic depletion of ROS.**

The wing pouch of each genotype (above) is complemented with an orthogonal apico-basal section (below); white lines indicate the level of the z-axis of the above images. (**A**) Anti-Wgn and Dcp1 of disc with genetically induced apoptosis using $sal^{E/Pv}$>*rpr* ($n = 12$). (**B**) Anti-Grnd and Dcp1 of disc with genetically induced apoptosis using $sal^{E/Pv}$>*rpr* ($n = 10$). (**C**) Anti-Wgn and Dcp1 of $sal^{E/Pv}$>*rpr, Sod1:Cat* ($n = 16$). (**D**) Anti-Grnd and Dcp1 of $sal^{E/Pv}$>*rpr, Sod1:Cat* ($n = 8$). (**E**) Anti-Wgn and Dcp1 of $sal^{E/Pv}$>*rpr, egr RNAi* ($n = 19$). (**F**) Anti-Grnd and Dcp1 of $sal^{E/Pv}$>*rpr, egr RNAi* ($n = 10$). Scale bars: 50 μm. (**G**) Box plots of the mean pixel intensities of Wgn inmunostaining of the genotypes $sal^{E/Pv}$>*rpr* ($n = 12$), $sal^{E/Pv}$>*rpr, Sod1::Cat* ($n = 16$), $sal^{E/Pv}$>*rpr, eiger RNAi* ($n = 19$). (**H**) Box plots of the mean pixel intensities of Grnd inmunostaining of the genotypes $sal^{E/Pv}$>*rpr* ($n = 10$), $sal^{E/Pv}$>*rpr, Sod1::Cat* ($n = 8$), $sal^{E/Pv}$>*rpr, eiger RNAi* ($n = 10$). Box plots show maximum–minimum range (whiskers), upper and lower quartiles (open rectangles), and median value (horizontal black line). Each dot corresponds to the mean pixel intensity of Wgn (**G**) or Grnd (**H**) measured on the anterior (A) or posterior (P) compartment (rectangles in the imaginal disc representation) of the imaginal discs analyzed. One-way ANOVA test was used for multiple comparisons between all groups: n.s. = not significant **$p = 0.0015$ and ****$p < 0.0001$, n.s. = not significant. Source data are available online for this figure.

system to induce apoptosis in the *sal*-central zone of the wing pouch and to knock down *wgn* or *grnd* in the wing pouch (henceforth *nub*>). Intact discs show low levels of phosphorylated p38 (Fig. 5C). In apoptosis-induced discs, we detected activation of p38 in the wing pouch surrounding the apoptotic cells (Fig. 5C,D), as previously described (Esteban-Collado et al, 2021; Santabárbara-Ruiz et al, 2015, 2019). Likewise, phosphorylation of p38 in cells near the apoptotic zone was found after knocking down *grnd* (Fig. 5E). However, after knocking down *wgn*, p38 phosphorylation in cells surrounding the apoptotic zone was strongly reduced (Fig. 5F). The accumulation of pyknotic nuclei after ectopic *rpr* expression indicates that apoptosis is not suppressed after knocking down *grnd* or *wgn*. As for the *ap/ci* drivers (Fig. 5B), the resulting $sal^{E/Pv}$>*rpr, nub*> *wgn RNAi* adult wings showed severe anomalies, indicating impaired regeneration (Fig. EV5B). These results demonstrate that *wgn*, but not *grnd*, is key for the activation of p38 signaling during the regenerative response to apoptosis. Remarkably, *wgn* inactivation in the same cells expressing *rpr* did not show significant defects, suggesting that *wgn* is required non-autonomously for regeneration (Fig. EV5C).

## Wgn and Grnd are phylogenetically divergent

Opposing roles between proteins of the TNFR superfamily suggest that they have an ancient origin and have followed divergent evolutionary paths. To track the differences observed between *grnd* and *wgn*, we decided to investigate the evolutionary origin of these two *Drosophila* genes. This involved searching for homologs in other animal lineages and performing maximum likelihood (ML) phylogenetic analyses using the amino acid sequences of the CRDs, the only conserved protein domain shared by all TNFRs (Figs. 6A and EV6A). We observed that *Drosophila* Grnd and Wgn CRD sequences were in separate branches of the tree, in two different, highly supported monophyletic groups that together included most of the TNFRs genes we had identified in arthropods (Table EV1, Fig. EV6B). The Wgn branch included genes from multiple arthropod groups, ranging from dipterans such as *Drosophila* to chelicerates, indicating that Wgn has a very ancient origin that predates the diversification of crown arthropods before the Cambrian, ~546 million years ago (mya) (Lozano-Fernandez et al, 2020) (Fig. 6B). In contrast, Grnd had a more restricted taxonomic distribution and was only present in pancrustacean species (insects plus crustaceans), suggesting that this gene family had a slightly younger origin (emerging at least 514 mya) (Wolfe et al, 2016) (Fig. 6B). Thus, these results showed that Grnd and Wgn are evolutionary distinct, suggesting they originated through an ancient duplication event of an ancestral TNFR in

pancrustaceans followed by a highly asymmetric evolution (Holland et al, 2017). Alternatively, *grnd* and *wgn* CRDs could have independent origins (through independent recruitment or exon shuffling with cysteine-rich regions from other gene families). Consistent with this, Grnd and Wgn clades were more closely related to certain TNFR gene families from deuterostome species than they were to each other (Figs. 6C and EV6C). However, in this case the bootstrap supports were too low to reliably assign orthology relationships, probably because CRD sequences are too divergent and short (less than 60 aa) to robustly establish evolutionary relationships between TNFR genes from distant animal phyla (i.e., human and insects) (Huang et al, 2008).

Therefore, we applied an alternative approach to assess similarities between the deuterostome TNFRs, Grnd, and Wgn by performing a PCA-based alignment (Fig. 6C) and a pairwise BLAST comparison (Fig. EV6C). In both analyses, Grnd CRD sequences clustered separate from all other TNFRs, while Wgn family members exhibited a higher degree of similarity with deuterostome TNFRs, clustering together with some of the CRD sequences from hemichordates (the acorn worm, *Saccoglossus kowalevskii*), non-vertebrate chordates (the European amphioxus, *Branchiostoma lanceolatum*), and vertebrates (*Homo sapiens*). Thus, Grnd would constitute a more divergent member of the TNFR superfamily than Wgn, a notion supported by the unique and derived pattern of cysteine residues that characterize Grnd CRDs, which contain 8 conserved cysteines instead of the 6 cysteines typically found in most TNFR families (Fig. EV6D) (Palmerini et al, 2021). These results reinforce the hypothesis of the repurposing of *Drosophila grnd* and *wgn* for different biological functions. The expansion and subfunctionalization of TNFR for apoptosis, survival, and the stress response has been suggested for other metazoan clades (Quistad and Traylor-Knowles, 2016).

## Discussion

In this work we have demonstrated that the conserved TNFR Wgn is activated by oxidative stress, accumulates near the source of ROS, and confers survival to cells even after knocking down Egr/TNFα. This function primarily involves p38 activity, likely via dTRAF1/Ask1, and is tightly linked to the oxidative stress induced after cell damage.

Mammalian TNFR pathways have pleiotropic functions as a result of complex regulatory mechanisms (Gough and Myles, 2020). Upon binding to TNFα, a core signaling complex is constructed on the cytoplasmic tail of the TNFRs. This signaling complex includes the TRAF adapter proteins as major signal transducers for the

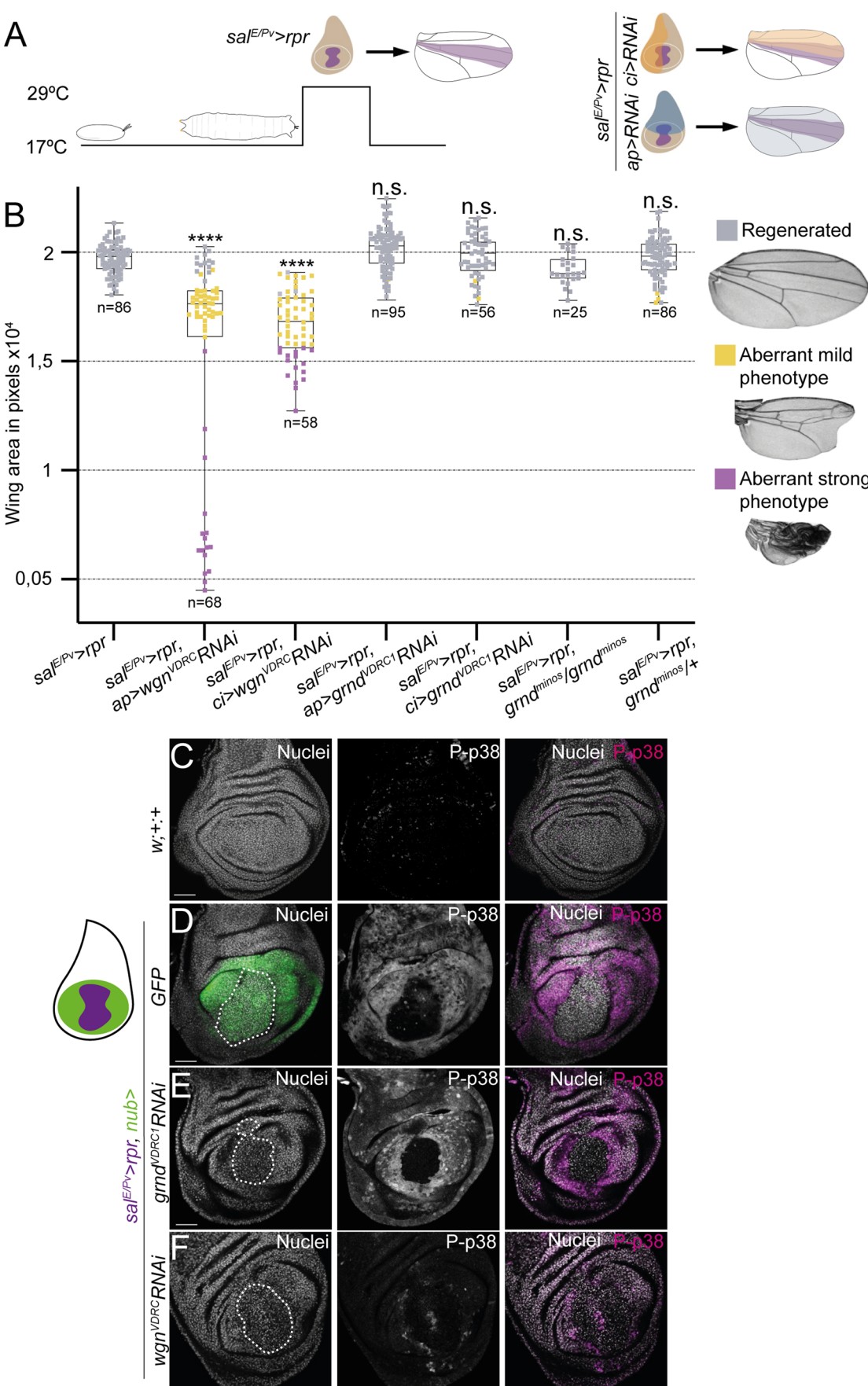

**Figure 5.  Knockdown of *wgn* but not *grnd* impairs p38-dependent regeneration.**

(A) Design for ectopic expression of transgenes and simultaneous apoptosis induction. Purple area in disc: apoptosis induced by *sal^{E/Pv}-LHG, LexO-rpr*. Light purple in adult wing: regenerated tissue. The Gal4/UAS activates RNAi in *ci*/anterior compartment (yellow), *ap*/dorsal compartment (blue). Adult wings were scored for complete regeneration of the missing zone. *sal^{E/Pv}-LHG* and *Gal4* are under the control of *tub-Gal80^{TS}*. Apoptosis was induced by shifting the temperature to 29 °C for 11 h. The larvae were transferred to 17 °C, where they regenerated and emerged into adults, in which wings were scored. (B) Regeneration assay. Box plot: Y-axis shows the average area in pixels of adult wings obtained after apoptosis in the *sal^{E/Pv}* region and the RNAi or mutant (genotype, X-axis). Each dot represents one wing; wild-type pattern/regenerated (gray), mild aberrant phenotype (yellow), and strong aberrant phenotype (purple). Box plots show maximum–minimum range (whiskers), upper and lower quartiles (open rectangles), and median value (horizontal black line), One-way ANOVA test was used for multiple comparisons between all groups: n.s. = not significant, ****$p < 0.0001$. (C–F) Activated p38 in regeneration assay of wing discs. Phosphorylated p38 (P-p38) in non-ablated disc (C; $n = 10$), in *sal^{E/Pv}-LHG, LexO-rpr* apoptosis and *nub-Gal* activation of GFP (D; ($n = 8$)), *UAS-grnd-RNAi* (E; ($n = 18$)), and *UAS-wgn-RNAi* (F; ($n = 12$)). White lines in the confocal images outline the *sal^{E/Pv}-LHG, LexO-rpr* dark area full of pyknotic nuclei of apoptotic cells. TP3 was used to stain the nuclei. Scale bar: 50 µm. Source data are available online for this figure.

TNFRs. In mammals and *Drosophila*, a range of biological functions, such as adaptive and innate immunity, embryonic development, and stress response, are mediated by TNFRs/TRAFs through the induction of cell survival, proliferation, differentiation, and/or cell death (Colombani and Andersen, 2023). Thus, TRAFs add complexity to the upstream TNFR and consequently to the signal transduction (Chung et al, 2002). In addition to TNFRs, TRAFs associate to MAPKKK such as Ask1 or Tak1, which will trigger the p38 or JNK MAPK pathways (Hoeflich et al, 1999; Nishitoh et al, 1998). In mammals, once TNFR1 is activated and binding has occurred with the core complex, typically comprised of TRAF2, TRAF5 and the receptor-interacting kinase RIPK1, post-translational modifications will determine its ability to activate p38 or JNK for survival and inflammation (Dostert et al, 2019). If these protein modifications are disrupted, the signaling complex triggers apoptosis (Dostert et al, 2019; Vince et al, 2009). Wgn signaling in *Drosophila* is also pleiotropic and its function in the cell will also depend on upstream and downstream signals. Wgn was first described as pro-apoptotic after binding to Egr/TNFα (Kanda et al, 2002), with dTRAF2 or dTRAF1 interacting with the TNFR (Geuking et al, 2005; Kauppila et al, 2003). However, the lack of the death domain, distinctive of many TNFRs, and the inability of *wgn* mutants to rescue the apoptotic phenotype generated by Egr/TNFα, suggests that apoptosis is not the main function of *wgn* (Andersen et al, 2015; Kanda et al, 2002). Indeed, Wgn is emerging as a pro-survival TNFR and, as shown in mammals, the combination of TRAFs or other adapter proteins in the C-terminal core complex could divert the pathway towards functions other than apoptosis. Indeed, our regeneration assay, where *Drosophila* Wgn-dTRAF1 has a survival function, as well as where Wgn-dTRAF3 suppresses lipolysis in the gut, are examples of a *Drosophila* TNFR acting independently of Egr/TNFα (Loudhaief et al, 2023). In contrast, Grnd/dTRAF2 drives the pathway towards JNK-driven apoptosis in a ligand-dependent manner, likely through the Tak1 MAPKKK (Andersen et al, 2015; Palmerini et al, 2021). Furthermore, Grnd displays nanomolar binding affinity for Egr that is three orders of magnitude higher than it is for Wgn, suggesting that canonical Egr signaling in *Drosophila* occurs predominantly through Grnd activation rather than through Wgn (Palmerini et al, 2021).

The TNFR superfamily has been described as containing a CRD on their N-terminal. Cysteine residues have active thiols that can be efficiently oxidized by ROS, a physiological mechanism to transmit external signals to the intracellular system (Gotoh and Cooper, 1998; Kamata et al, 2005; Zhang et al, 2001) and subsequently alter protein structure, interaction with partners, and subcellular localization (Sies et al, 2017). Moreover, human TNFR1/2 can suffer oxidative stress-induced self-association due to modifications

of the CRD, resulting in ligand-independent signaling (Ozsoy et al, 2008). Therefore, we speculate that, similarly to mammals, the oxidation of Wgn could promote its self-association and influence the partner preferences of the core signaling complex, i.e., Wgn-dTRAF1, in a ligand-independent manner. In addition, dTRAF1 has been described to be a positive interaction partner of Ask1 (Kuranaga et al, 2002). Hence, we propose that Wgn-dTRAF1-Ask1 is a signaling module activated upon oxidative stress to ensure survival in cells that will be involved in the regenerative response. Moreover, we recently demonstrated that Ask1 requires the activity of the nutrient-dependent Pi3K/Akt signal to divert Ask1 function to p38 phosphorylation in cell survival and regeneration (Esteban-Collado et al, 2021). Thus, we conclude that it is not only Pi3K/Akt, but also Wgn, that will be necessary for leading Ask1 to induce a p38 response to apoptosis.

Here, we propose a model for the targets that respond to ROS upon cell damage. Cells that have been damaged, either by injuries or apoptosis, produce ROS, normally of mitochondrial origin (Murphy, 2009). ROS spreads from damaged or dying cells to the nearby healthy cells, acting as early signals for tissue recovery in different organisms such as flatworms and mammals (Gauron et al, 2013; Rampon et al, 2018). In our model, the ROS-dependent post-translational modifications will primarily target three branches that converge with Ask1 (Fig. 7). First, we have the thioredoxin bound to the inactive Ask1 signaling complex, which upon oxidation will be dissociated and allow the active Ask1 to interact with TRAFs (Matsuzawa, 2016; Nishida et al, 2017; Sakauchi et al, 2017; Shiizaki et al, 2013; Tobiume et al, 2001). Second, there is the insulin/Pi3K/Akt signaling pathway that will phosphorylate Ask1 to reroute it for survival and ultimately phosphorylation of p38 (Esteban-Collado et al, 2021; Kim et al, 2001; Santabárbara-Ruiz et al, 2019). Third, because we have found that the Wgn/dTRAF1/Ask1 axis is involved in survival as a stress response, we hypothesize that Wgn oxidation could be a signal that contributes to the interaction of dTRAF1 with the Ask1 core signaling complex (Noguchi et al, 2005). Thus, the divergent role of Grnd and Wgn is driven not only by their different affinity for the ligand Egr/TNFα, but also by the ROS-dependent activation of Wgn. We have shown here that Wgn accumulation near the damaged zone can be reverted after ROS depletion, indicating that ROS produced by dying cells is involved in the Wgn response after damage. Lineage experiments have shown that these cells near the damaged zone are responsible for most of the regenerated epithelium (Bosch et al, 2008; Repiso et al, 2013). However, we cannot rule out that the accumulation of Wgn near the affected area responds not only to a reorganization of vesicles, but also to a transcriptional response. Indeed, RNAseq of regenerating imaginal discs has shown that *wgn* is transcriptionally

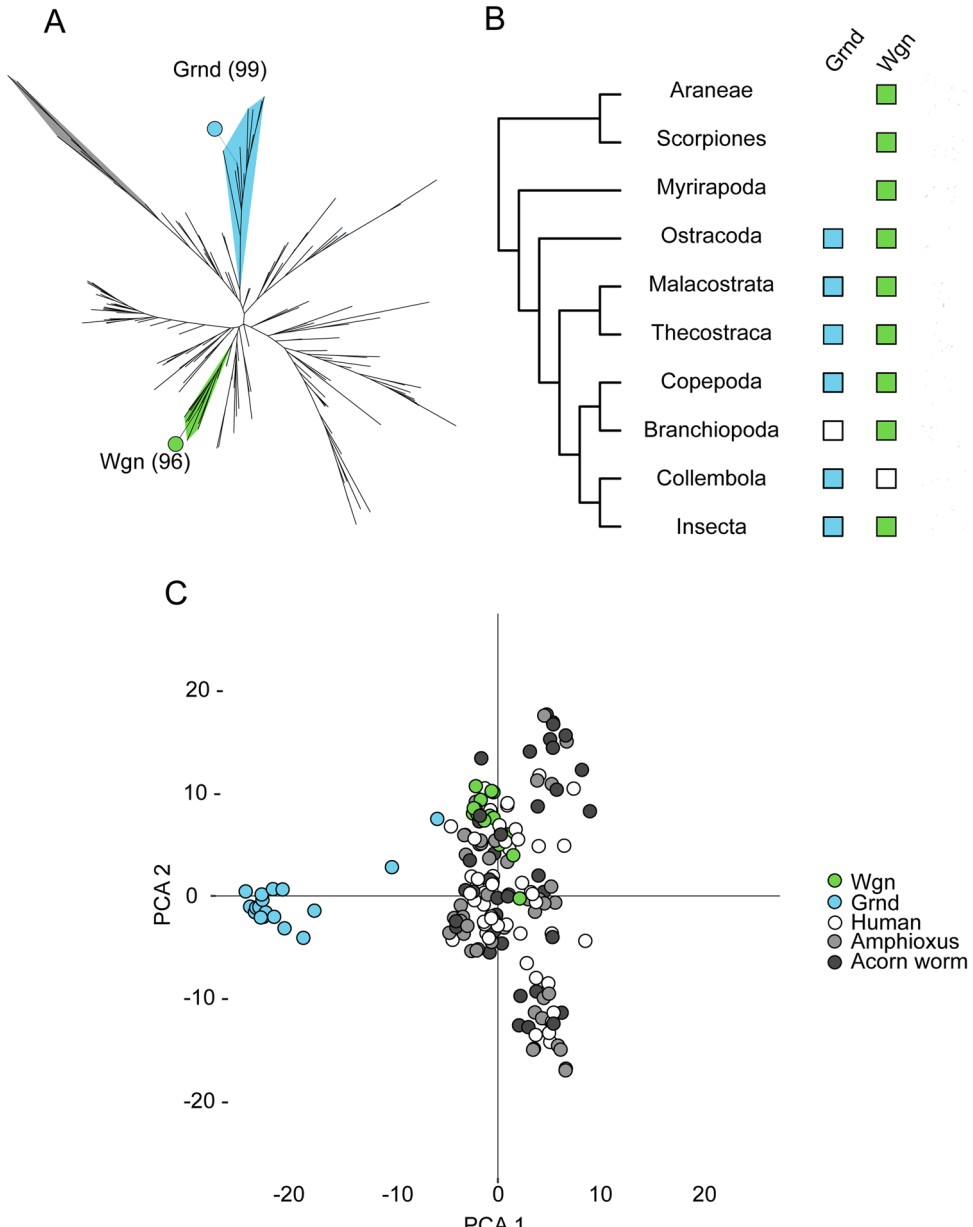

**Figure 6. Evolution of Wgn and Grnd.**

(A) ML phylogenetic tree of the CRDs of Wgn, Grnd, and deuterostome TNFRs. *Drosophila melanogaster* Wgn and Grnd are indicated by solid green and light blue circles, and their corresponding monophyletic clades are colored accordingly. The outgroup branch, containing the cysteine-rich sequences of LIM proteins, is indicated in gray. (B) Wgn and Grnd presence/absence in different arthropod lineages. The presence of the two gene families is indicated by solid squares; white squares indicate putative gene losses. Divergence times of crown arthropods and crown pancrustaceans are indicated at the corresponding nodes of the arthropod tree. (C) PCA-based alignment of the CRDs of Wgn, Grnd, and Deuterostome TNFRs.

upregulated in the earliest phase of regeneration (Vizcaya-Molina et al, 2018).

We observed a very high degree of divergence between Grnd and Wgn, showing that both gene families are of extremely ancient origin and are found across pancrustacean and arthropod lineages, respectively. This taxonomic distribution suggests that Grnd originated through a duplication and fast divergence at the base of the pancrustacean phylogenetic tree. These results match the subfunctionalization observed in both genes, with Grnd having a

pro-apoptotic function and Wgn promoting cell survival. The amplification and subfunctionalization of TNFRs has also been observed in different lineages for adaptation to biotic or abiotic stress (Quistad and Traylor-Knowles, 2016).

Remarkably, we observed that there is also a very high sequence divergence between the CRDs of Grnd and those of all the other TNFR families we studied. By contrast, Wgn CRDs were much more canonical, and we found that they clustered together with the CRDs from deuterostome TNFRs, including those from humans.

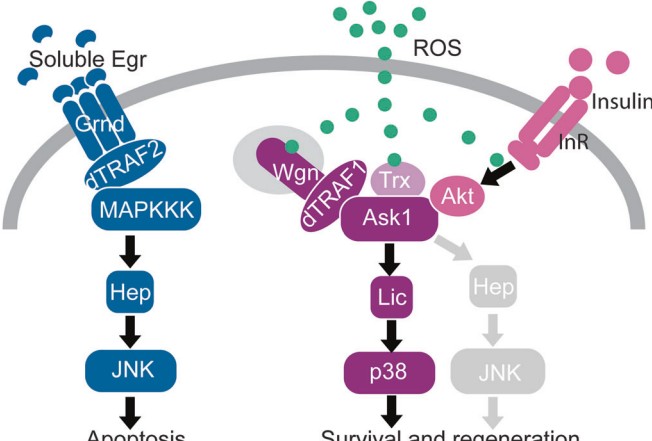

**Figure 7. Model for ROS-dependent Wgn recruitment for survival and regeneration.**

ROS are required for the oxidation of thioredoxin (Trx) to dissociate from Ask1. ROS are required for the phosphorylation of Akt, which in turn phosphorylates Ask1 to divert its function towards survival and regeneration. ROS are required for Wgn recruitment in the Ask1/p38 axis. Phosphorylation of Ask1 by Akt results in tolerable levels of p38 and JNK (perhaps very low, gray) activity and the avoidance of apoptosis. In contrast to Wgn, Egr-Grnd interaction triggers JNK-dependent apoptosis.

We speculate that the subcellular location of Wgn and Grnd may contribute to the different functions of both receptors. Grnd is more exposed at the apical side of the plasma membrane, which makes this receptor more accessible for ligand interactions (Palmerini et al, 2021) to transduce a JNK-dependent apoptotic signal (Moreno et al, 2002b; Shlevkov and Morata, 2012; Tobiume et al, 2001). Wgn, embedded in cytoplasmic vesicles, is less accessible to the ligand and could be more restricted to being activated by local sources of signaling molecules, such as ROS. In contrast to initial reports (Kanda et al, 2002; Kauppila et al, 2003), *wgn* loss of function does not rescue Egr-induced apoptosis in the *Drosophila* eye (Andersen et al, 2015), which supports our observation in the wing that Wgn is not required for Egr-induced apoptosis. Instead, Egr-induced apoptosis generates ROS, which target intracellular Wgn to foster a cell survival program in cells adjacent to the apoptotic zone.

# Methods

## *Drosophila* strains

Animals were reared on standard fly food. The *sal^{E/Pv}-LHG* and *lexO-rpr Drosophila* strains have been previously described (Santabárbara-Ruiz et al, 2015). Other strains were: *UAS-egr^{weak}* (Moreno et al, 2002a), *UAS-lic^{WT1.1}* (Terriente-Félix et al, 2017), *UAS-hep^{WT}* (Uhlirova and Bohmann, 2006), *sal^{E/Pv}-Gal4* (Barrio and de Celis, 2004), *ci-Gal4* (Martin and Morata, 2006), *UAS-eiger-RNAi* (Kuranaga et al, 2002). The *eiger-lacZ* strain was provided by K. Basler. The *w^{118};+;+* strain was used as a control. These strains are described in FlyBase: *ap-Gal4, nub-Gal4*. From the Bloomington Drosophila Stock Center (BDSC): *ptc-Gal4* (RRID:BDSC_2017), *tub-Gal80^{TS}* (RRID:BDSC_7017), *UAS-rpr* (RRID:BDSC_92781), *UAS-Sod1.a*

(RRID:BDSC_24754), *UAS-Cat.a* (RRID:BDSC_24621), *UAS-p35* (RRID:BDSC_5072), *UAS-wgn-RNAi* (TRiP, RRID:BDSC_55275), *UAS-Ask1-RNAi* (RRID:BDSC_35331 ; TRIP1 in figures), *UAS-GFP* (RRID:BDSC_4776), *UAS-Tak1-RNAi* (RRID:BDSC_53377), *UAS-dTRAF2-RNAi (6)* (RRID:BDSC_33931), *UAS-dTRAF1-RNAi (4)* (RRID:BDSC_55226), *UAS-Ask1* (RRID:BDSC_32464; TRIP2 in figures). From the Vienna Drosophila Resource Center (VDRC): *Egr-GFP* (VDRC 318615), *UAS-dTRAF1-RNAi* (VDRC 110766), *UAS-grnd-RNAi* (CG10176, VDRC 104538; VDRC1 in figures), *UAS-grnd-RNAi* (CG10176, VDRC43454; VDRC2 in figures), *UAS-wgn-RNAi* (GD9152, VDRC 9152), *UAS-dTRAF2-RNAi* (VDRC 110266), *UAS-p38-RNAi* (CG7393, VDRC 330146), *UAS-Tak1-RNAi* (CG18492; VDRC330457).

## Immunolocalization

The primary antibodies used in this work were against phospho-p38 from rabbit (1:50, (Cell Signaling Technology Cat# 9211, RRID:AB_331641)), rabbit anti-cleaved Death Caspase-1 (Dcp1) (1:200, (Cell Signaling Technology Cat# 9578, RRID:AB_2721060)), ß-galactosidase (mouse 1:1000, (Promega Cat# Z3783, RRID:AB_430878)), Wengen (mouse 1:100, gift from K. Basler), Grindelwald (guinea pig 1:500, gift from Pierre Leopold). The fluorescently labeled secondary antibodies were from Life Technologies, all used 1:200 in 0.3% Triton-PBS: goat anti-mouse Alexa Fluor 488 (Thermo Fisher Scientific Cat# A-11001, RRID:AB_2534069), donkey anti-rabbit Alexa Fluor 568 (Thermo Fisher Scientific Cat# A10042, RRID:AB_2534017), and goat anti-guinea pig Alexa Fluor 555 (Thermo Fisher Scientific Cat# A-21435, RRID:AB_2535856).

Wing imaginal discs dissected from late third instar larvae in 1×PBS (pH 7.4) were fixed in 4% paraformaldehyde (Electron Microscopy Sciences, Cat# 15710) in PBS for 40 min at room temperature, washed in PBS 0.3% Triton X-100 (PBT), blocked for 2 h in PBT containing 2% BSA, and incubated overnight with primary antibodies at 4 °C. The next day, the discs were washed and incubated with secondary antibodies for 2 h at room temperature. After washing, nucleic acid staining was performed by incubating the discs for 10–15 min with the nuclear marker TO-PRO-3, 1 mM (TP-3, Thermo Fisher Scientific Cat. # T3605) or DAPI, 10 mM (Thermo Fisher Scientific, Cat.# D21490). The discs were mounted in SlowFade Diamond Antifade Mountant (Thermo Fisher Scientific, Cat.# S36967).

## Image acquisition

For the confocal images, a Zeiss LSM880 and a Leica SPE confocal laser scanning microscopes were used. Images were analyzed and processed using FIJI. A Leica DMBL microscope was used for taking pictures of the adult wings.

## Quantification of the Wgn and Grnd response to damage

Stacks of images corresponding to Wgn and Grnd localization in these three genotypes were used for quantification: (1) *sal^{E/Pv}>rpr*; (2) *sal^{E/Pv}>rpr, Sod1:Cat*; and (3) *sal^{E/Pv}>rpr, egr RNAi*. Mean pixel intensities were collected from raw images taken with the same confocal settings. Rectangular ROIs were traced to measure mean pixel intensity in two regions of the same disc (A: anterior, P: posterior), both outside the death zone.

## Statistical analysis and cell death area ratio calculation

To calculate the cell death area ratio, we used FIJI. First, we generated a Z-projection of the whole stack for the anti-Dcp1 channel using the Max intensity projection. Then, we applied the remove outliers tool (radius 2, threshold 10). Subsequently we thresholded the image with the MaxEntropy Threshold to create a binary image and automatically determine the area in pixels of the dead zone with the analyze particles tool. The data was normalized by dividing the area of Dcp1-positive cells by the total area of the imaginal disc. As a result, the cell death area ratio was obtained.

In Figs. 1 and 3, the data are in mean ± SD. To make statistical comparisons, we used a one-way analysis of variance (ANOVA) followed by Tukey's post hoc test to make pair comparisons between each group using GraphPad. Significance is indicated in the figures only when $p < 0.05$, as follows: $*p < 0.05$, $**p < 0.01$, $***p < 0.001$, and $****p < 0.0001$.

In Fig. EV6B we used a two-tailed Student's t-test to make the statistical comparisons using GraphPad ($****p < 0.0001$).

## Gal4/UAS/Gal80ts for eiger^weak activation in the wing imaginal disc

The Gal4/UAS transactivation system was temporarily controlled by the *tub-Gal80^TS* thermo-sensitive *Gal4*-repressor. *Egr/TNFα* was ectopically expressed using the *UAS-eiger^weak* construct (Moreno et al, 2002b), which results in reduced *egr* activity and therefore low levels of apoptosis. The expression of the transgene was controlled by the thermo-sensitive Gal4 repressor *tub-Gal80^TS*.

*Drosophila* of the desired genotype were cultured to lay eggs for 24 h at 17 °C. Conditions for experiments in Figs. 1 and 3: Embryos were kept at 17 °C until the 8th day (192 h) after egg laying to prevent *UAS-eiger^weak* expression. The larvae were subsequently transferred to 29 °C for 16 h and then the imaginal discs from wandering larvae were dissected and processed for staining and immunofluorescence studies.

Conditions for experiments in Fig. 4: To enhance the production of *egr*-dependent apoptotic cells, embryos were kept at 17 °C until the 7th day (168 h) after egg laying to prevent *UAS-eiger^weak* expression. The larvae were subsequently transferred to 29 °C for 24 h and then the imaginal discs from wandering larvae were dissected and processed for staining and immunofluorescence studies.

## ROS detection ex vivo

All experiments for ROS detection were done in living conditions. To detect the presence of ROS, we used the MitoSOX reagent (Thermo Fisher Scientific, Cat. #M36008), which is an indicator of oxidative stress of mitochondrial origin in living cells. Third instar discs were dissected in Schneider's medium immediately after cell death or injury and incubated in agitation for 15 min in medium containing 5 µM MitoSOX reagent, followed by three washes in Schneider's culture medium (Sigma-Aldrich S0146). The samples were protected from light throughout the experiment. They were then mounted using culture medium supplemented with 1 µM TO-PRO-3 for nucleic acid staining. As in vivo TO-PRO-3 only enters dying and dead cells, we used it to distinguish dead cells from living cells.

## Genetic ablation and the dual Gal4/LexA transactivation system

For adult wing regeneration analysis, we used a dual Gal4 and lexA transactivation system, as described previously (Esteban-Collado et al, 2021; Santabárbara-Ruiz et al, 2019). Briefly, the lexA/lexO system was used for genetic ablation after induction of apoptosis, and the Gal4/UAS system for activating the desired transgene. The *UAS-wgn-RNAi* and *UAS-grnd-RNAi* transgenes were activated under the control of *ci-Gal4, ap-Gal4,* or *nub-Gal4* (Figs. 5 and EV5).

The genetic ablation system used to study regeneration was *sal^{E/Pv}-LHG, tub-Gal80^{TS}, lexO-rpr*. Apoptosis was induced during larval stages in the wing-specific *sal^{E/Pv}* domain using the *Gal80*-repressible transactivator system LHG (LexA-Hinge-Gal4 activation domain), a modified form of the lexA/lexO system (*-LHG>lexO-rpr*) (Yagi et al, 2010) (Fig. 5A).

Flies of the desired genotype were allowed to lay eggs during 6 h at 17 °C. Eggs were allowed to develop on standard fly food. Genetic ablation was achieved after temperature shifts from 17 °C to 29 °C of 11 h in vials containing the synchronized larvae.

The ability of the wing disc to regenerate after genetic ablation has been associated with the induction of a developmental delay (Colombani et al, 2012; Garelli et al, 2012; Jaszczak et al, 2015; Katsuyama et al, 2015; Smith-Bolton et al, 2009). All genotypes analyzed in Fig. 6 showed a similar developmental delay of 1–2 days (at 17 °C) after genetic ablation in comparison with the animals of the same genotype in which no genetic ablation was induced, i.e., developed continuously at 17 °C (Fig. EV5A). After the adults emerged, the wings were dissected, and regeneration was analyzed.

To test the capacity to regenerate, we used adult wings from *sal^{E/Pv}>rpr* individuals, in which patterning defects can be easily scored. Flies were fixed in glycerol:ethanol (1:2) for 24 h and the wings were dissected in water and then washed with ethanol. They were then mounted in 6:5 lactic acid:ethanol and analyzed and imaged under a Leica microscope.

The areas of the mounted wings were outlined and scored using FIJI. In addition, we divided the wings into three categories depending on the severity of the defects in terms of pattern and number of absent veins and interveins: regenerated (normal pattern), mild aberrant phenotype (1–2 veins missing), and strong aberrant phenotype (>3 veins missing or more aberrant phenotype).

The use of the knock down RNAi transgenes alone did not affect vein pattern or wing size (Fig. EV5A). Also, flies carrying the constructs shown in Fig. 6 but raised constantly at 17 °C to maintain *tub-Gal80^{TS}* activity and block transgene (*UAS or lexO*) expression did not show defects in wing size or pattern (Fig. EV5A).

This double transactivation system was also used to dissect imaginal discs after genetic ablation and *grnd* or *wgn* down-regulation and score the effects on phospho-p38 localization (Fig. 5). In these experiments, we used the same conditions as those for adult wings and the *nub-Gal4* driver was selected because its expression domain surrounds the *sal^{E/Pv}* zone.

## Characterization of Grnd and Wgn gene family evolution

Protein sequences of TNFRs were researched using the blastp and tblastn software from the online Blast server (https://blast.ncbi.nlm.nih.gov/Blast.cgi), restricting by taxonomy to specifically search in the genomes

and proteomes of those animal lineages of particular relevance for Grnd and Wgn evolution. *Drosophila melanogaster* Grnd and Wgn were used as initial queries. After that initial search, a second one was performed using the retrieved protein hits as new queries. The same procedure was used in two slow-evolving deuterostome lineages (a cephalochordate, the European amphioxus *Branchiostoma lanceolatum*, and a hemichordate, the acorn worm *Saccoglossus kowalevskii*), using *Homo sapiens* TNFRs as initial queries.

For the phylogenetic analyses of TNFRs, protein sequences were aligned with the MAFFT software (Katoh et al, 2019) and the resulting alignments were trimmed using Aliview to discard spuriously aligned regions (Larsson, 2014). Phylogenetic trees were built using IQ-Tree (Nguyen et al, 2015), testing the tree with UFBoot (bootstrap = 103) (Hoang et al, 2018) and an approximate Bayes test for single branch testing. The model used was selected using ModelFinder with BIC as the criteria (Kalyaanamoorthy et al, 2017). The trees were visualized using ITOL (Letunic and Bork, 2021).

Two phylogenetic trees were built. The first one was done using only the CRDs from all the arthropod and deuterostome TNFRs retrieved in our searches, except for a few highly divergent sequences from Ostracoda and Thecostraca (1 from *Cyprideis torosa* and 2 from *Amphibalanus amphitrite*). In this tree, all arthropod TNFRs were assigned with high support to either Grnd or Wgn monophyletic groups, except in the case of a TNFR from the spider *Argiope bruennichi*, which grouped together with some deuterostome sequences with very low support (Figs. 6A and EV6A). The second tree was built using the full protein sequences of Wgn and Grnd from different arthropods, together with the few remaining arthropod TNFR sequences that could not be classified in the previous tree (the 4 sequences from *C. torosa*, *A. amphitrite*, and *A. bruennichi*), which in this case were successfully assigned as Grnd or Wgn orthologs (Fig. EV6B).

For the CRD tree, the corresponding protein domains were extracted using Hmmer 3.4 (http://hmmer.org/). LIM domains, which are also cysteine-rich domains unrelated to those of TNFRs, from the human proteins ISL1, LIMCH1, and FHL1 were used as an outgroup. The model selected for this tree was PMB + G4.

The Grnd and Wgn tree was built with the model VT + I + G4 + F. Three TNFRs from deuterostomes were used as outgroups, TNFR6L from *B. lanceolatum* (CAH1240663.1), TNFR11L from *S. kowalevskii* (XP_006819214.1), and TNFR1B from *H. sapiens*.

A PCA-based alignment of all the CRDs used in the previous phylogenetic analyses was built using the Jalview software (Waterhouse et al, 2009) and represented with R. Also, an all versus all comparison was performed using blastp. The percent identity was normalized with the total length of the query. The results were represented using R (github.com/rlbarter/superheat).

## Data availability

This study includes no data deposited in external repositories.

The source data of this paper are collected in the following database record: biostudies:S-SCDT-10_1038-S44318-024-00155-9.

## Peer review information

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

## Acknowledgements

The authors would like to thank Dr. Manel Bosch from the Optical Microscopy Unit of the CCiT od the Universitat de Barcelona for his assistance. We also thank Marco Milán and Lidia Pérez for sharing the wild-type form of *licorne* (*UAS-lic*$^{WT1.1}$). We also thank Paula Santa Bárbara Ruiz for her preliminary observations. This research was funded by the Spanish Ministerio de Ciencia, Innovación y Universidades (PID2021-123300NB-I00) and by the Agència de Gestió d'Ajuts Universitaris i de Recerca of the Generalitat de Catalunya (2021SGR00293) to FS, MC, and JEC, and by Spanish Ministerio de Ciencia, Innovación y Universidades(PID2021-128728NB-I00 and CNS2022-136105) to NM and MFM.

## Author contributions

**José Esteban-Collado**: Conceptualization; Data curation; Formal analysis; Validation; Investigation; Visualization; Methodology; Writing—original draft; Writing—review and editing. **Mar Fernández-Mañas**: Formal analysis; Investigation. **Manuel Fernández-Moreno**: Data curation; Software; Formal analysis; Validation; Investigation; Methodology. **Ignacio Maeso**: Conceptualization; Formal analysis; Supervision; Validation; Investigation. **Montserrat Corominas**: Conceptualization; Formal analysis; Supervision; Funding acquisition; Investigation; Visualization; Writing—original draft; Project administration. **Florenci Serras**: Conceptualization; Resources; Formal analysis; Supervision; Funding acquisition; Validation; Investigation; Visualization; Methodology; Writing—original draft; Project administration; Writing—review and editing.

Source data underlying figure panels in this paper may have individual authorship assigned. Where available, figure panel/source data authorship is listed in the following database record: biostudies:S-SCDT-10_1038-S44318-024-00155-9.

## Disclosure and competing interests statement

The authors declare no competing interests.

# Expanded View Figures

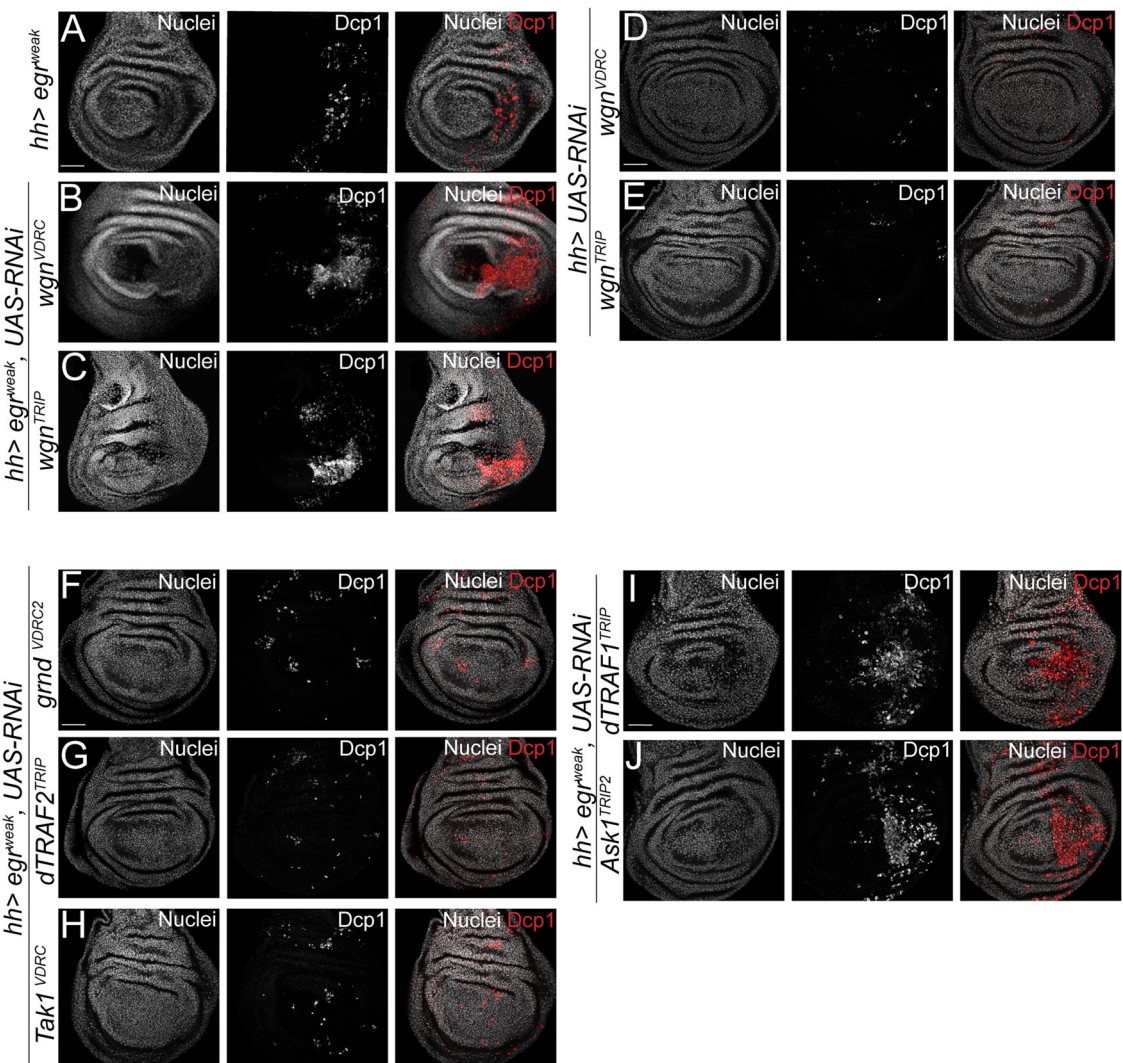

**Figure EV1. Dcp1-positive cells in different RNAi strains expressed under *hh-Gal4*.**

(A) Control *egr^weak^* overexpression only. (B) *egr^weak^* and *wgn RNAi* from VDRC. (C) *egr^weak^* overexpression and *wgn RNAi*, TRiP strain. (D) Control expression of *wgn RNAi*, VDRC alone. (E) Control expression of *wgn RNAi*, VDRC alone. (F) *egr^weak^* and *grnd RNAi*, TRiP. (G) *egr^weak^* and *sTRAF2 RNAi*, TRiP. (H) *egr^weak^* and *Tak1 RNAi*, VDRC. (I) *egr^weak^* and *dTRAF1 RNAi*, TRiP. (J) *egr^weak^* and *Ask1 RNAi*, VDRC. TP3 was used to stain nuclei. Scale bar: 50 μm. Source data are available online for this figure.

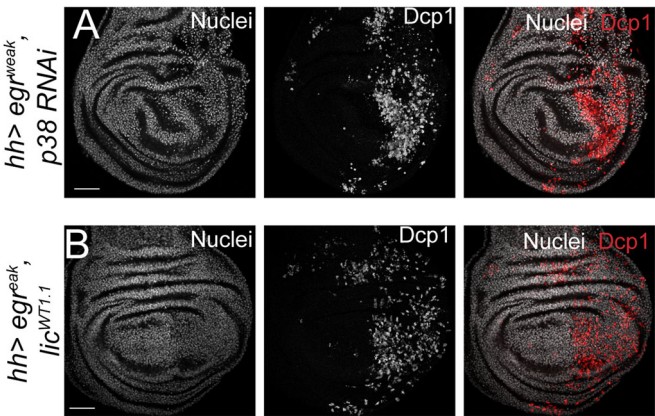

**Figure EV2. Reduction or increase of p38 activity results in an increase of apoptosis in *egr^weak^* tissue.**

(A) *UAS-p38-RNAi* and *UAS egr^weak^* transgenes were co-expressed under *hh-Gal4*. (B) *UAS-lic^WT1.1^* and *UAS egr^weak^* transgenes were co-expressed under *hh-Gal4*. Nuclei staining was done with TO-PRO-3 and apoptosis with the caspase Dcp-1. Scale bar: 50 μm. Source data are available online for this figure.

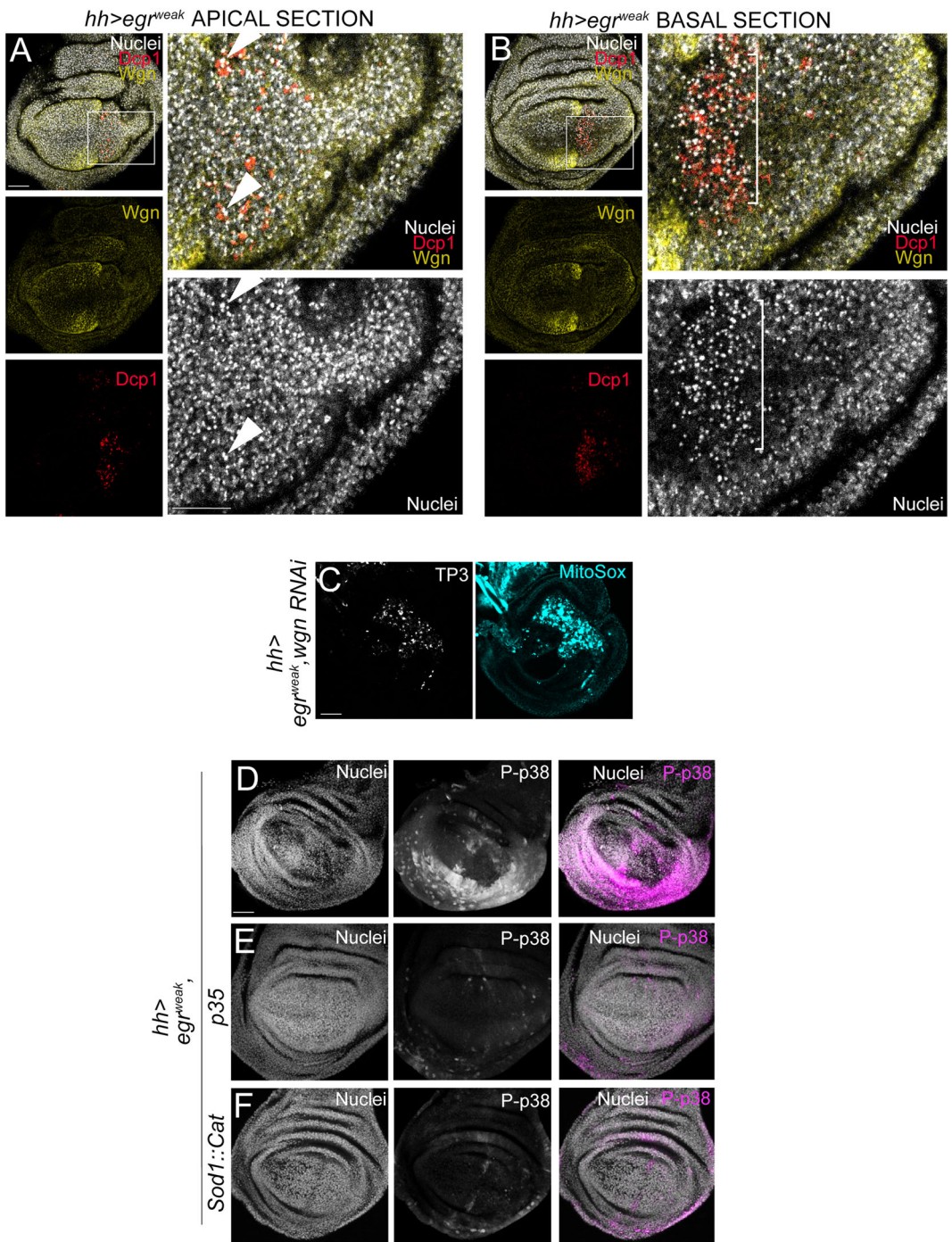

**Figure EV3. p38 activation depends on ROS generated by apoptotic *egr* cells.**

(A, B) *Hh-Gal4 UAS-egr^weak* wing disc to show pyknotic nuclei positive for the cleaved caspase Dcp1 in apical-basal polarity. (A) Plane close to the apical side of the epithelium with a high magnification of the square zone. Note that pyknotic nuclei coincide with Dcp1 cells (e.g., arrows). (B) Plane close to the basal side of the epithelium with a high magnification of the square zone. Note that pyknotic nuclei coincide with Dcp1 cells (e.g., bracket), and that the concentration of apoptotic cells is more abundant in the basal area. (C) Discs stained ex vivo after co-expression of *egr^weak* and *wgn RNAi*. Cell death (TP3), ROS of mitochondrial origin (MitoSOX). (D–F) Phosphorylation of p38 after ectopic expression of *egr^weak*. (D) The transgene *egr^weak* was activated in the posterior compartment (*hh-Gal4*); in this optical section there is an accumulation of pyknotic nuclei, typical of apoptotic cells in the center of the posterior wing pouch. Phospho-p38 is found in cells surrounding the apoptotic cells. (E) The expression of the apoptosis inhibitor *p35*, and (F) the ROS scavengers *Sod1::Cat* concomitantly with *egr^weak* result in a strong reduction of phospho-p38. Scale bar: 50 μm.

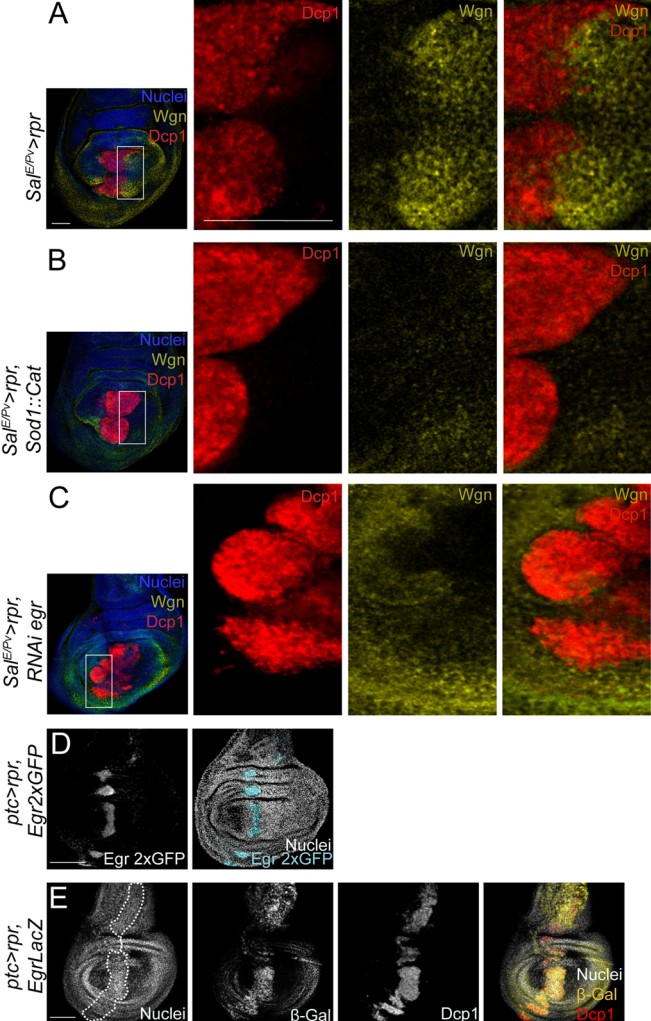

**Figure EV4. Wgn response to apoptotic ROS occurs independently of Egr/TNFα.**

(A–C) High magnification of the interphase between dying cells and responding tissue corresponding to Fig. 5. (A) Anti-Wgn and Dcp1 of disc with genetically induced apoptosis using *sal^{E/Pv}>rpr*. (B) Anti-Wgn and Dcp1 of *sal^{E/Pv}>rpr, Sod1:Cat*. (C) Anti-Wgn and Dcp1 of *sal^{E/Pv}>rpr, egrRNAi* (*n* = 19). (D, E) Apoptosis genetically induced in *ptc>rpr* in two different Egr reporter backgrounds; (D) *Egr2xGFP* reporter and (E) *EgrLacZ* detected by anti-ß-Gal antibody. The yellow zone in the merged image shows co-localization of ß-Gal-positive cells and Dcp1-positive cells (dead cells). The dotted lines outline pyknotic nuclei of apoptotic cells. TP3 was used to stain the nuclei. Scale bar: 50 μm.

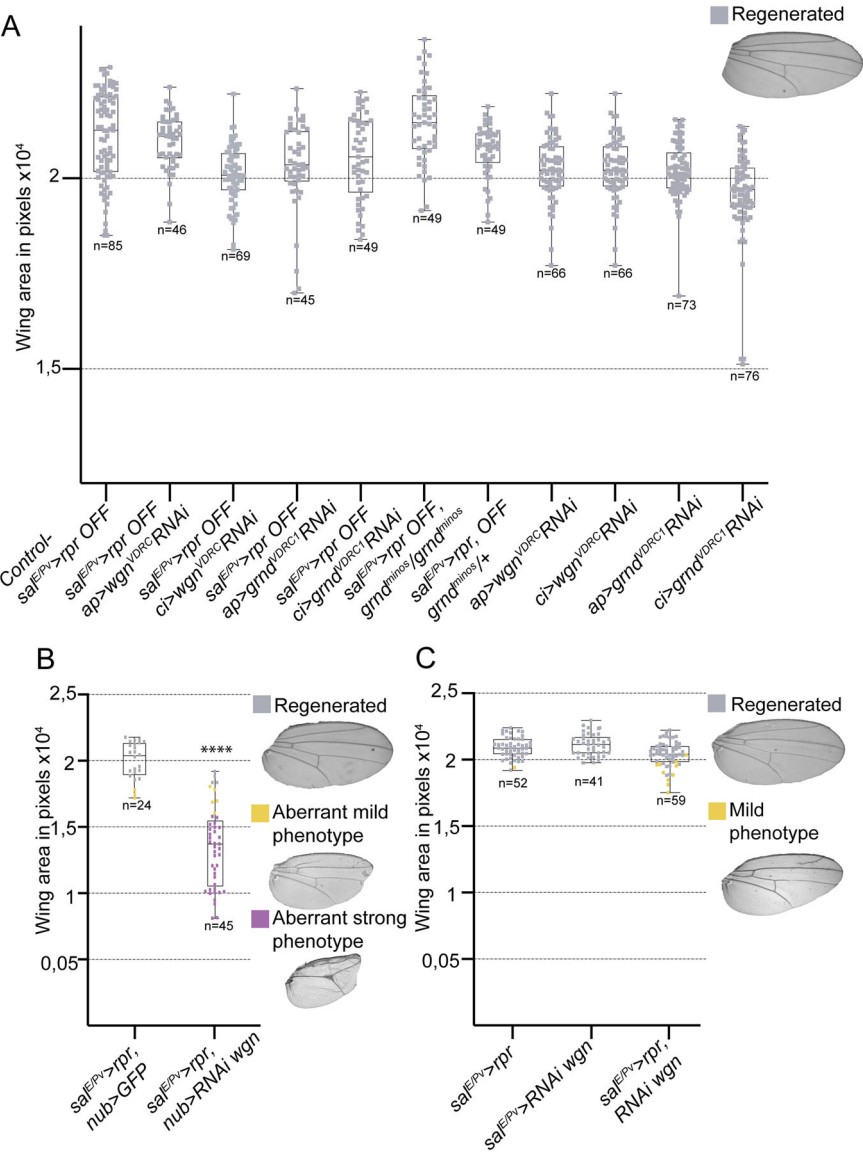

**Figure EV5. Control for regeneration assay.**

(**A**) Box plot: Y-axis shows the average area in pixels of adult wings obtained from controls kept at 17 °C, with no cell death induction (*sal^{E/Pv}-LHG,LexO-rpr* OFF) and no expression of the transgenes. It also shows the average area in pixels from adult wings after the sole expression of the *RNAi* or mutant background (genotypes indicated in the X-axis). Each dot represents one wing: wild-type pattern (gray). One-way ANOVA test was used for multiple comparisons between all groups. (**B**) Regeneration assay. Box plot: Y-axis shows the average area in pixels of adult wings obtained after apoptosis in the *sal^{E/Pv}* region and the *wgn RNAi* in the *nub* zone (as in Fig. 6F, G). Each dot represents one wing; wild-type pattern/regenerated (gray), mild aberrant phenotype (yellow), and strong aberrant phenotype (purple). T-Student test was used for comparison of the means between the two groups: ****$p < 0.0001$. (**C**) Regeneration assay. Box plot: Y-axis shows the average area in pixels of adult wings obtained after apoptosis in *sal^{E/Pv}>rpr*, in *sal^{E/Pv}> wgn RNAi* and in *sal^{E/Pv}>rpr + wgn RNAi*. Each dot represents one wing; wild-type pattern/regenerated (gray), mild aberrant phenotype (yellow), One-way ANOVA test was used for multiple comparisons between all groups. Graphs box plots show maximum–minimum range (whiskers), upper and lower quartiles (open rectangles), and median value (horizontal black line).

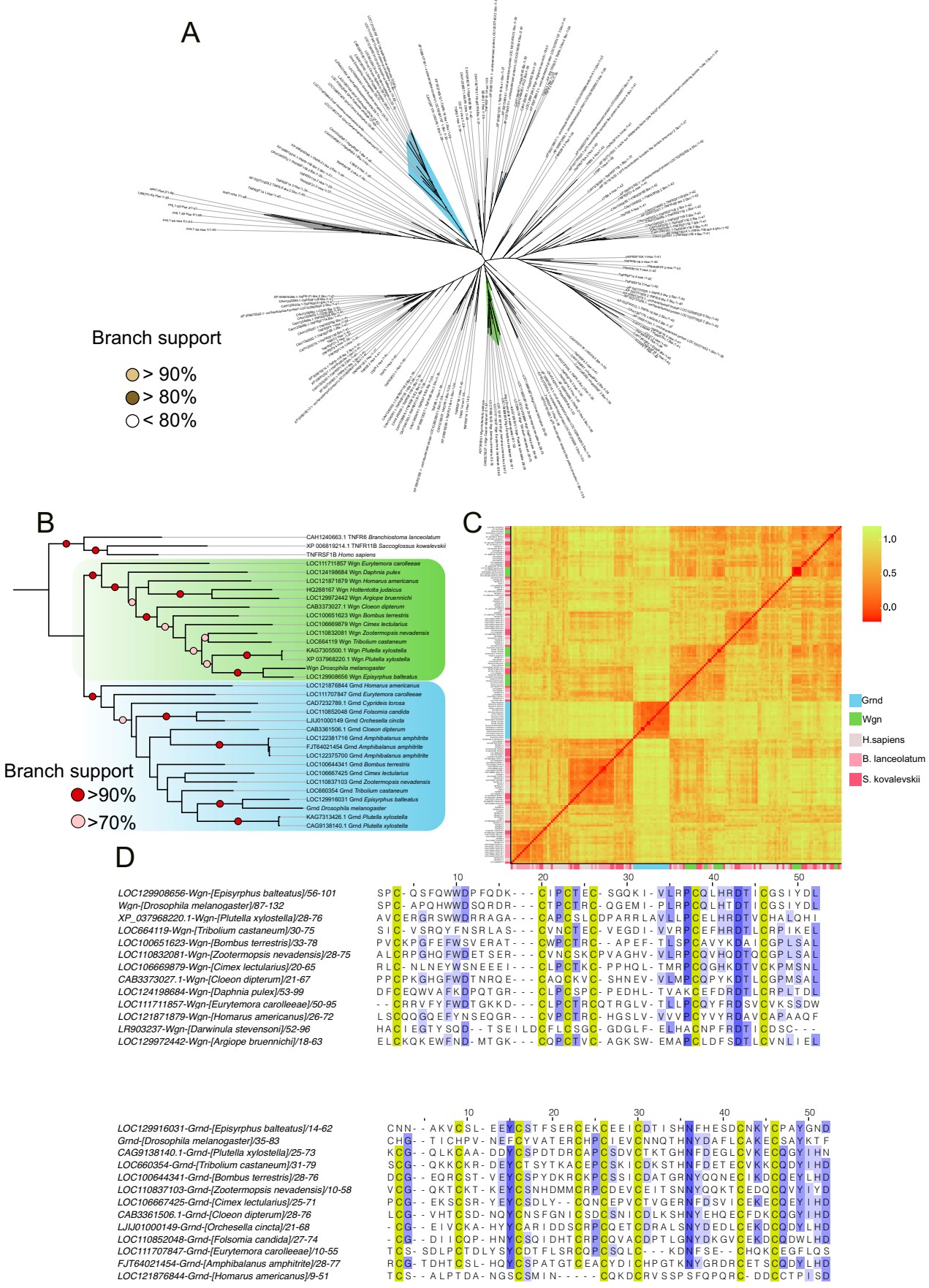

◄   **Figure EV6.   Phylogeny and evolutionary conservation of CDRs from Wgn and Grnd.**

(**A**) Full version, including all the species names and sequence accession numbers of the ML phylogenetic tree of the CRDs of Wgn, Grnd, and deuterostome TNFRs featured in Fig. 6A. *B. lanceolatum*, *S. kowalevskii*, and *H. sapiens* are abbreviated as Bla, Sko and Hsa, respectively. (**B**) ML phylogenetic tree of all identified arthropod proteins, using full-length protein sequences and three deuterostome TNFRs as outgroups, *B. lanceolatum* TNFR6L (CAH1240663.1), *S. kowalevskii* TNFR11L (XP_006819214.1), and *H. sapiens* TNFR1B. (**C**) Pairwise comparison of Wgn, Grnd, and deuterostome CRDs. (**D**) CRD alignments of some representative Wgn and Grnd proteins.

