## [Peer Review File · The EMBO Journal]

Reactive oxygen species activate the *Drosophila* TNF receptor Wengen for damage-induced regeneration

José Esteban-Collado, Mar Fernández-Mañas, Manuel Fernández-Moreno, Ignacio Maeso, Montserrat Corominas, and Florenci Serras

Corresponding author: Florenci Serras (fserras@ub.edu)

Review Timeline:

Transfer from Review Commons:	31st Jan 24
Editorial Decision:	5th Feb 24
Revision Received:	19th Apr 24
Editorial Decision:	23rd May 24
Revision Received:	5th Jun 24
Accepted:	12th Jun 24

Editor: Ieva Gailite

Transaction Report:

This manuscript was transferred to The EMBO Journal following peer review at Review Commons.

Review #1

1. Evidence, reproducibility and clarity:

Evidence, reproducibility and clarity (Required)

Tumor necrosis factor (TNF)- α stands out as a remarkably conserved pro-inflammatory cytokine that plays crucial roles in immunity, tissue repair, and cellular homeostasis. The *Drosophila* TNF-TNF receptor (TNFR) system, known for its simplicity, combined with a versatile genetic toolkit, has been instrumental in unraveling the intricate mechanisms governing both the physiological and pathological functions mediated by TNF. Recently, the fly TNFR Wengen has been described to have ligand independent functions in maintaining tissue homeostasis and tracheal remodeling. The current manuscript describes a novel TNF/Egr-independent function of Wengen in regulating tissue regeneration in imaginal discs. The authors identify both the upstream regulator (ROS) and the downstream signaling pathway through Ask1/p38 MAPK. The data presented are solid and support an interesting model where ROS emanating from damaged tissue triggers Wgn-dependent signaling in adjacent cells to promote regeneration. Few points could be addressed:

Major points:

- The result in Fig1.H is somehow surprising. How does the overexpression of Egr induce caspase activation in the absence of its receptor Grnd? Is it possible that the loss of function of Wengen on its own has a phenotype? If so, that would suggest that Wgn in addition to its role in regeneration might be implicated in pro-survival processes in homeostatic conditions?
- In Fig.6, it would be relevant to include wengen inactivation within the domain where rpr is expressed to show that wengen is not required autonomously for regeneration (*sal>rpr + wgn RNAi*). What is the phenotype of the adult wing of *sal-lexA>rpr + nub-gal4 >wgn RNAi* animals?
- Is the overexpression of Wengen sufficient to increase tissue regeneration?

Minor points:

- In fig.1I, it is surprising that knockdown of neither Grnd nor dTRAF2 significantly affects Egr-induced apoptosis
- The ability of the wing disc to regenerate has been associated with the induction of a developmental delay mediated by Dilp8. Are the authors observing this developmental delay in the case of *sal-lexA>rpr + Ap-gal4 >wgn RNAi* or *sal-lexA>rpr + Ci-Gal4>wgn RNAi*
- It is possible to test the induction of apoptosis in a wgn null mutant background to see if the phenotype is even stronger than the one observed with RNAi (the wgn RNAi is induced at the same time as egr or rpr overexpression).
- The investigation of the evolutionary origin of TNFR in *Drosophila* included in Fig.2 is cutting a bit the flow of the results.
- The authors could explain in more details the double transactivation system for non-fly specialists.

- It can be interesting to include and/or discuss these few references:

PLoS Genet. 2019 Aug; 15(8): e1008133.

PLoS Genet. 2022 Dec 5;18(12):e1010533.

FEBS Lett. 2023 Oct;597(19):2416-2432.

Nat Commun. 2020 Jul 20;11(1):3631.

Curr Biol. 2016 Mar 7;26(5):575-84.

2. Significance:

Significance (Required)

The understanding of the mechanistic interplay between TNFR in integrating TNF-dependent and independent signals to stimulate distinct downstream responses lays the foundation for investigating whether these insights can be generalized to other members within the TNFR superfamily in all organisms. This work is relevant for a large audience of researchers working in the field of inflammation and TNFR.

3. How much time do you estimate the authors will need to complete the suggested revisions:

Estimated time to Complete Revisions (Required)

(Decision Recommendation)

Between 3 and 6 months

Yes

Review #2

1. Evidence, reproducibility and clarity:

Evidence, reproducibility and clarity (Required)

The *Drosophila* genome encodes a single TNF α ortholog, *eiger* (*egr*), and two TNF Receptors (TNFR), *wengen* (*wgn*) and *grindelwald* (*grnd*). While *egr* overexpression can cause apoptosis, the authors here report that *wgn* and *grnd* have opposing roles in cell death and survival. Specifically, the authors show evidence that *grnd* promotes cell death in response to *egr* expression, while *wgn* promotes cell survival through the p38 MAP Kinase pathway. They further show that apoptotic cells have high levels of ROS, which activates the *wgn*-p38 axis for tissue regeneration, independent of *egr*.

Overall, the manuscript is well written. At the same time, there are some technical concerns and missing controls that need to be addressed. Below are a few specific comments for the authors' consideration:

Major Comments

1. The authors find that *wgn* RNAi enhances *hh>egr*-induced apoptosis. They validate the results with two independent RNAi lines in Figure S1. However, Figure S1 is missing a control without *wgn* RNAi, and therefore, the results are difficult to assess.
2. Are the two independent *wgn* RNAi lines targeting different regions of the coding sequence?
3. Aside from *wgn*, other RNAi experiments are not validated through independent RNAi lines. I suggest expanding the Supplemental Figures to reproduce a few key findings with independent RNAi lines.
4. In Figure 1E, the authors show that *wgn* RNAi enhances cell death caused by *hh>egr*. What is missing here is a *wgn* RNAi control without *hh>egr*. Is there any cell death caused by the loss of *wgn* alone (without *hh>egr*)?
5. If *wgn* RNAi causes some degree of cell death, is the observed effect with *hh>egr* a significant genetic interaction, or merely additive?
6. Is the *wgn*-p38 pathway sufficient to block *egr* induced cell death? The authors could test this by having *hh>egr* in the *licT1.1* background. The authors have a more complex experiment in Figure 3, where *licT1.1* is introduced into the *hh>egr*, *wgn* RNAi background. However, testing the effect of *licT1.1* without *wgn* would establish a more direct relationship between *egr* and *wgn*-p38.
7. In Figure 4, the authors show that *egr* expression induces ROS and performs anti-oxidant experiments. This part could be strengthened if they show that the ROS sensor signal disappears after *Sod::Cat* expression.
8. How effective is *egr* RNAi? In Figure 5E, F, the authors knock down *egr* and obtain negative results. Based on this, the authors argue that *Wgn* localization occurs through an *egr*-independent mechanism. Drawing strong conclusions based on a negative result with *egr* RNAi is not a good

practice since one cannot rule out residual *egr* activity that mediates the effect. I suggest either finding ways to completely abolish *egr* function, or tone down the conclusion.

9. Figure 6 shows that *wgn* RNAi aggravates the reaper phenotype. What's missing is a control that expresses *wgn* RNAi but not reaper.

2. Significance:

Significance (Required)

There is now a detailed understanding of mammalian TNFRs, which play pro-apoptotic and non-apoptotic roles depending upon the context. Previous studies had also reported that TNFR1 respond to ROS. By comparison, our understandings of the two TNFRs in *Drosophila* remain rudimentary. The two receptors have different loss-of-function phenotypes, some of which may be independent of *egr* signaling. The major significance of this work is in delineating the distinct behaviors of the two *Drosophila* TNFRs, centering around their pro-apoptotic, or pro-survival properties.

Audience: This study will draw the interest of *Drosophila* geneticists, those interested in Reactive Oxygen Species and cell death, and evolutionary biologists.

3. How much time do you estimate the authors will need to complete the suggested revisions:

Estimated time to Complete Revisions (Required)

(Decision Recommendation)

Between 1 and 3 months

Yes

Review #3

1. Evidence, reproducibility and clarity:

Evidence, reproducibility and clarity (Required)

Summary:

TNF receptors have a broad range of possible function, from inducing apoptosis to promoting cell survival and proliferation. How this works is not completely understood. *Drosophila* has two TNF receptors, Wengen and Grindelwald. This manuscript nicely shows that Grindelwald is pro-apoptotic while Wengen promotes cell survival and proliferation. Strikingly, if TNF is expressed in *Drosophila* tissue, knockdown of the receptor Wengen leads to elevated levels of apoptosis, clearly showing its cell-protective function. Interestingly, the authors find that Wengen is activated by ROS produced by neighboring dying cells - regardless of whether they are dying due to TNF signaling or not - and that Wengen then activates p38 downstream to mediate a regenerative response.

Major comments:

Overall the conclusions are interesting, clear and convincing. The data are of very good quality. I only have a few minor comments below.

Minor comments:

1. Fig 6C-E would need a control disc without induced apoptosis (ie wildtype disc) stained for phospho-p38 as a baseline comparison. This is important to judge the significance of the remaining phospho-p38 in panel E where wgn is knocked down. The authors write " However, after knocking down wgn, phosphorylated p38 in the wing pouch surrounding the apoptotic cells was abolished (Fig. 6E)." Depending on the amount of phospho-p38 in control discs, this may need to be rephrased to "strongly reduced" instead of "abolished".
2. This sentence in the Intro needs fixing because TNF α doesn't transduce the signal from TNFR to Ask1 since it's upstream of TNFR:
"TNF α can transduce the TRAF-mediated signal from TNFR to Ask1 to modulate its activity (Hoefflich et al., 1999; Nishitoh et al., 1998, p. 0; Obsil & Obsilova, 2017; Shiizaki et al., 2013)."
3. In the results section, the authors start by ectopically overexpressing Eiger. Are there conditions where Eiger expression is induced in the wing? If yes, it would be helpful for the reader to mention that this system with the genetically GAL4-induced expression of Eiger aims to phenocopy one of these conditions.
4. Fig 2C is not very self-explanatory: it is worth writing out what Hsa (*H. sapiens*), Bla and Sco

stand for (there is plenty of space).

5. This sentence is confusing:

"...Wgn localization were due to ROS or to the expression of egr, we used RNAi to knock down egr in the apoptotic cells and found that reduced Egr/TNF α had no effect on Wgn localization (Fig. 5E, 5F)."

The authors may want to specify that Wgn is still accumulated even without Egr. ("No effect" is unclear).

2. Significance:

Significance (Required)

This manuscript makes several important discoveries:

1. it clearly shows that one TNF receptor, Grindelwald, is mainly pro-apoptotic, while the other, Wengen, is mainly pro-survival. This provides a mechanistic explanation for the dual role of the TNF, Eiger.
2. It discovers that Wengen is activated by ROS. In fact, since Wengen binds TNF with an affinity that is several orders of magnitude lower than Grindelwald, and since Wengen is not even located at the cell membrane, these data call into question whether Wengen is a TNF receptor, or a ROS receptor? Could the authors comment on this? Could it be that the results obtained in the past showing that Wengen is activated by TNF were actually due to TNF inducing apoptosis, leading to production of ROS, leading to activation of Wengen?
3. It was previously shown that damage, for instance in the fly intestine, induces production of ROS, which then activates p38, leading to a proliferative/regenerative response. This manuscript provides a missing mechanistic link, showing that the ROS activates Wengen, which in turn activates p38. This thereby completes the mechanistic chain of events from damage to the regenerative response.

Hence, overall, this is a very interesting study. It will be of interest for a broad audience of people studying TNF signaling, stress signaling and stress response, tissue damage and repair, and regeneration.

My expertise: Drosophila, growth, signaling

3. How much time do you estimate the authors will need to complete the suggested revisions:

Estimated time to Complete Revisions (Required)

(Decision Recommendation)

Less than 1 month

No

Review #4

1. Evidence, reproducibility and clarity:

Evidence, reproducibility and clarity (Required)

The study by Florenci Serras and colleagues presents compelling evidence highlighting distinct functions of the two *Drosophila* TNFRs, Wengen (Wgn) and Grindewald (Grnd) in developing imaginal epithelia. The study shows that while Grnd-dTraf2-Tak1 module controls apoptosis in Eiger (Egr)-dependent manner, Wgn-dTraf1-Ask1 promotes survival likely via p38 signaling independent of Egr. Their phylogenetic analysis underscores the ancient origin of both receptors while revealing their divergent evolutionary path, manifested by markedly different CRD sequences. Moreover, Wgn shows higher degree of similarity to mammalian TNFRs. Using functional genetics and confocal imaging of immunostained wing imaginal discs, the authors confirm the differential localization of Wgn and Grnd in imaginal cells, consistent with several recent studies. Interestingly, they demonstrate distinct responses between Grnd that is internalized upon Eiger[weak] overexpression from the plasma membrane, and Wgn, which cytoplasmic levels decrease in Eiger[weak] OE cells but become enriched in neighboring wild type cells. The non-autonomous accumulation of Wgn required ROS but not Eiger production by dying cells. Finally, employing an elegant double-driver *lexA/lexO* and *Gal4/UAS* system, enabling independent gene manipulation in specific domains of the wing imaginal discs, the authors established the essential role of Wgn, but not Grnd, in the regenerative response to apoptosis, including the occurrence of phosphorylated p38.

****Major comments:****

The conclusion regarding the protective role of p38 in response to Egr[weak] should be supported by a p38 knockdown experiment. Are phospho-p38 levels increased in cells expressing Egr[weak]?

In Figure 4C it appears that the Dcp-1 positive cells move apically rather than basally. Including nuclear staining would be very informative allowing assessment of tissue morphology. The authors focus on the pouch region of the wing imaginal disc, where phenotypes are strong and obvious. However, the hh-Gal4 driver also affects posterior cells in the hinge and notum, where the effects of Eiger[weak] overexpression seem weaker (e.g., minimal to no MitoSox signal in hinge and notum posterior cells). Could the authors explain this observation?

In Figure 5, the cells expressing Rpr appeared to be pulled/extruded basally as expected. It would be beneficial to quantify Wgn and Grnd signals along cross-sections and provide higher magnification images of domain boundaries to illustrate differences in TNFR localization and levels.

The micrographs for Grnd Figure 5B,D, F capture substantial signal from the peripodial epithelium where the sale/Pv> driver is likely not active?

The non-autonomous induction of Wgn seems stronger when facing dying Rpr overexpressing cells simultaneously depleted of Eiger compared to RprOE alone. Should this be a reproducible, could the authors discuss potential reason for this observation?

Figure 6 C-E. Does WgnRNAi potentiates and GrndRNAi suppress Rpr-induced apoptosis similarly to their effects when knocked down in Eiger[weak]OE cells? The activation of p38 following sale/Pv>rpr-mediated ablation as shown by immunostaining is noteworthy. While loss Grnd knockdown leads to phospho-p38 signal enrichment around the rpr-expressing cells, WgnRNAi results in reduced phospho-p38 signal in the wing pouch but also beyond the nub-expression domain. Do sale/Pv>rpr nub>WgnRNAi cells still generate ROS? Are ROS responsible for the long-range signaling and p38 activation, referring to authors' previous work Santabá rbara-Ruiz et al., 2019, PLoS Genet 15(1): e1007926. <https://doi.org/10.1371/journal.pgen.1007926>, Figure 5G?

****Minor comments:****

I propose rephrasing the description of "UAS-Egr[weak] transgene, a strain that produces a reduced Egr/TNF α activity". It could imply a loss of function strain rather than a transgene that causes mild/moderate Egr overexpression.

I recommend the authors to revise the charts for improved clarity in genotype representation. For example, in Figure 1I, the label "control-GFP" might be misleading. It would be beneficial to specify that "control" refers to Eiger[weak] alone with other manipulations being done simultaneously with Eiger[weak] overexpression. Additionally, considering that individuals with color blindness may struggle to differentiate between red and green colors, I strongly suggest using a color-blind-friendly palette, especially in Figure 4A, C, G, and Figure 4A, C, E."

Providing detailed information regarding the reagents used in the study, such as Catalogue Numbers or RRIDs, is beneficial for enhancing reproducibility.

2. Significance:

Significance (Required)

This is a very solid study that uncovers unique roles of *Drosophila* TNFRs in regulating imaginal cell behaviors crucial for tissue regeneration. It expands our knowledge on processes controlled by TNFR-mediated signaling, highlighting the potential for ligand-independent regulation. The study nicely complements recent findings by several laboratories (Letizia et al., 2023; Loudhaief et al.,

2023; Palmerini et al., 2021). Beyond its contribution to fundamental biology, the study has biomedical implication for regenerative medicine. It emphasizes the necessity of balancing TNFR activities, downstream signaling and their dependence on ligands, providing important insights for the development of receptor agonists or antagonists. The findings are relevant to audience interested in developmental and regenerative biology, gene evolution.

**Strengths:* functional genetics revealing distinctive roles for the two TNFRs in *Drosophila* and their dependency on ligand in the paradigm of tissue regeneration.

**Limitations:* quality of the imaging - higher magnification images and quantification would enhance the study.

The summarizing model may contain excessive speculations that lack support from the data or references to the existing literature.

3. How much time do you estimate the authors will need to complete the suggested revisions:

Estimated time to Complete Revisions (Required)

(Decision Recommendation)

Between 3 and 6 months

Yes

Revision Plan

Manuscript number: #RC-2023-02257

Corresponding author(s): Florenci SERRAS

1. General Statements [optional]

All four reviewers have positive comments on the paper. We totally agree with their comments, and proposed controls and experiments. Most of them are already introduced in the present text and several new figures added, as we had the controls/experiments proposed. Few others are now being done and we hope to have the complete set of experiments ready in 2-3 months.

2. Description of the planned revisions

Insert here a point-by-point reply that explains what revisions, additional experimentations and analyses are planned to address the points raised by the referees.

Reviewer #1

Most comments of this reviewer have already been done and included in the transferred manuscript, except for part of the first comment:

1.1 b. Is it possible that the loss of function of Wgn on its own has a phenotype? If so, that would suggest that Wgn in addition to its role in regeneration might be implicated in pro-survival processes in homeostatic conditions?

This issue is very important to understand the differential role of Wgn and Grnd. First of all, Wengen knock out (*wgn*^{KO}; Andersen et al., 2015) is viable in homozygosis. However, in this paper we have focused on inducible mutants. Therefore, we have now crossed the flies to get the genotype *hh-Gal UAS RNAi wgn* and we will check for apoptotic phenotype, as suggested. This will take us **few weeks** of work.

Reviewer #2

Most comments have been already carried out and included in the transferred manuscript, except these ones:

2.3. Aside from wgn, other RNAi experiments are not validated through independent RNAi lines. I suggest expanding the Supplemental Figures to reproduce a few key findings with independent RNAi lines.

We have recently received a set of independent RNAi line to repeat the experiments for Traf1, Traf2, Ask1 from Bloomington Stock Center. And We did not do it before mainly because we wanted to focus on *wgn* and *grnd*. However, we agree with the Reviewer 2 and we will do the experiments. Another RNAi from VDRC for *grnd* and *Tak1* have been ordered. These experiments will take about **2 months** from the crosses to the analysis of results (some flies still to arrive, and many crosses will be done at 17°C).

Revision Plan

2.4. In Figure 1E, the authors show that *wgn* RNAi enhances cell death caused by *hh>egr*. What is missing here is a *wgn* RNAi control without *hh>egr*. Is there any cell death caused by the loss of *wgn* alone (without *hh>egr*)?

This control is now in progress. Expected to have it complete in **2 weeks**.

2.5. If *wgn* RNAi causes some degree of cell death, is the observed effect with *hh>egr* a significant genetic interaction, or merely additive?

The result from the previous comment will help us to respond this point.

2.6. Is the *wgn*-p38 pathway sufficient to block *egr* induced cell death? The authors could test this by having *hh>egr* in the *licT1.1* background. The authors have a more complex experiment in Figure 3, where *licT1.1* is introduced into the *hh>egr*, *wgn* RNAi background. However, testing the effect of *licT1.1* without *wgn* would establish a more direct relationship between *egr* and *wgn*-p38.

We have set the crosses for the experiment *hh>egr* and *licT1.1* as suggested. The results will be included in the new version of the manuscript. **1 month**.

Reviewer #3

All comments already carried out and included in the transferred manuscript. See next sections.

Reviewer #4

Major comments:

4.3 In Figure 5, the cells expressing *Rpr* appeared to be pulled/extruded basally as expected. It would be beneficial to quantify *Wgn* and *Grnd* signals along cross-sections and provide higher magnification images of domain boundaries to illustrate differences in TNFR localization and levels.

The micrographs for *Grnd* Figure 5B,D, F capture substantial signal from the peripodial epithelium where the *salE/Pv>* driver is likely not active?

We will do a thorough quantification of high-resolution stacks of images and include higher magnification of the analyzed stacks. To this aim, we need some more weeks to collect the images of each genotype, processed and quantify them. We propose to do have this work done in **two months**.

4.4 The non-autonomous induction of *Wgn* seems stronger when facing dying *Rpr* overexpressing cells simultaneously depleted of *Eiger* compared to *Rpr* OE alone. Should this be a reproducible, could the authors discuss potential reason for this observation?

It is difficult to respond this question, without quantification. The quantification suggested in the previous point, will allow us to state if *Wgn* is more accumulated in *rpr +egr* than *rpr* alone. Therefore, the previous point will tell us if there are significant differences and if, so it will help us to discuss it.

Timing: The entire plan can be executed in 2-3 month.

3. Description of the revisions that have already been incorporated in the transferred manuscript

Please insert a point-by-point reply describing the revisions that were already carried out and included in the transferred manuscript. If no revisions have been carried out yet, please leave this section empty.

Reviewer #1

1.1 a- *The result in Fig1.H is somehow surprising. How does the overexpression of Egr induce caspase activation in the absence of its receptor Grnd?*

The results of Fig. 1H, in which *egr+grndRNAi+wgnRNAi* results in high apoptosis indicates that *wgn* down regulation compromises survival even in the absence of *grnd*. The reviewer correctly points that “*How does the overexpression of Egr induce caspase activation in the absence of its receptor Grnd?*”.

There is evidence that Eiger is involved in the regulation of the pro-apoptotic gene *head involution defective (hid)* in primordial germ cells (Maezawa 2009 Dev. Growth Differ., 51 (4) (2009), pp. 453-461) and in the elimination of damaged neurons during development (Shklover et al., 2015). Moreover, Eiger is necessary for HID stabilization and regulates HID-induced apoptosis independently of JNK signaling (Shklover et al., 2015). Therefore, in our discs *egr* activation in the absence of *grnd* and *wgn* can still result in apoptosis because of the absence of *wgn*'s survival signal and, presumably, activation of *hid*.

We have introduced this issue in the text as:

“To check for epistasis between *grnd* and *wgn*, we activated *hh> egrweak* and knocked down both TNFRs. We found high levels of cell death compared to *wgn RNAi* alone (Fig. 1H and 1I), which demonstrates that *wgn* down-regulation is dominant over *grnd*. This is surprising as it is generally assumed that Egr interacts with Grnd to induce apoptosis via JNK, which in turn activates the proapoptotic gene *hid* (Andersen et al., 2015; Diwanji & Bergmann, 2020; Fogarty et al., 2016; Igaki et al., 2002; Moreno, Yan, et al., 2002; Sanchez et al., 2019; Shlevkov & Morata, 2012). Interestingly, Egr is necessary for HID stabilization and can regulate HID-induced apoptosis independently of JNK (Shklover et al., 2015). Therefore, cells *egrweak* that downregulate *grnd* and *wgn* can still be eliminated because the lack of both Wgn-survival signal and the pro-apoptotic Grnd/JNK signal could result in an alternative pathway of apoptosis.”

1.2- *In Fig.6, it would be relevant to include wengen inactivation within the domain where rpr is expressed to show that wengen is not required autonomously for regeneration (sal>rpr + wgn RNAi). What is the phenotype of the adult wing of sal-lexA>rpr + nub-gal4 >wgn RNAi animals.?*

Revision Plan

We have already added a new figure (Fig. S4C) containing this data. As shown, both *wgn*RNAi alone and *wgn* RNAi + *rpr* do not show relevant anomalies and regenerate normally. Therefore, we conclude that *wgn* is not autonomously required for regeneration.

The adult wings *sal-lexA>rpr + nub-gal4 >wgn* RNA result in a strong aberration, as regeneration is inhibited. This experiment has been also added in another figure (Fig. S4B) it is done.

1.4 Minor- In fig. 1I, it is surprising that knockdown of neither Grnd nor dTRAF2 significantly affects Egr-induced apoptosis

After applying a One-Way ANOVA test to compare all the groups against all the groups in fig. 1B no significant differences were detected between Control and RNAi *grnd* or RNAi *dTRAF2* ($p > 0,05$). But if we apply a Student's T test, which is less restrictive, we obtain, indeed, significant differences:

Control vs. RNAi *grnd* $p = 9,48 \times 10^{-7}$

Control vs. RNAi *dTRAF2* $p = 2,47 \times 10^{-7}$

We have now added in the text:

"Note that when *egr*weak cells downregulated *dTRAF2* or *grnd* the cell death area ratio is slightly lower than *egr*weak alone (Fig. 1I), confirming that *dTRAF2* and *Grnd* contribute to apoptosis in *egr*weak cells."

1.5 Minor The ability of the wing disc to regenerate has been associated with the induction of a developmental delay mediated by Dilp8. Are the authors observing this developmental delay is the case of sal-lexA>rpr + Ap-gal4 >wgn RNAi or sal-lexA>rpr + Ci-Gal4>wgn RNAi

The developmental delay due to *Dilp8* has been observed by many laboratories, indeed. The question of the reviewer is relevant because if there is no delay in pupariation, regeneration could be compromised not because regeneration has been affected but because after pupariation regeneration is impeded.

However, delay in pupariation has been found in our experiments. Usually for 11hrs of heat shock (to induce apoptosis) we found 1-2 days of delay.

We have added the following text:

"The ability of the wing disc to regenerate after genetic ablation has been associated with the induction of a developmental delay (Colombani et al., 2012; Garelli et al., 2012; Jaszczak et al., 2015; Katsuyama et al., 2015; Smith-Bolton et al., 2009). All genotypes analyzed in figure 6 showed a similar developmental delay of 1-2 days (at 17°C) after genetic ablation in comparison to the animals of the same genotype in which no genetic ablation was induced, i.e. developed continuously at 17°C (Fig. S4A). After the adults emerged, the wings were dissected, and regeneration was analyzed."

1.7 Minor - The investigation of the evolutionary origin of TNFR in drosophila included in Fig.2 is cutting a bit the flow of the results.

Revision Plan

The evolutionary origin starts now with a sentence that can smoothen the flow and few changes in that paragraph have been made:

“Opposing roles between proteins of the TNFR superfamily suggests that they have an ancient origin and have followed divergent evolutionary paths. To track the differences observed between *grnd* and *wgn*, we decided to investigate the evolutionary origin of these two *Drosophila* genes.”

1.8 Minor *The authors could explain in more details the double transactivation system for non-fly specialists.*

The entire section has been re-written in Material and Methods.

1.9 Minor - *It can be interesting to include and/or discuss these few references:*

PLoS Genet. 2019 Aug; 15(8): e1008133.

PLoS Genet. 2022 Dec 5;18(12):e1010533.

FEBS Lett. 2023 Oct;597(19):2416-2432.

Curr Biol. 2016 Mar 7;26(5):575-84.

Nat Commun. 2020 Jul 20;11(1):3631.

All these references, and few others, have been introduced in the text.

Reviewer #2

2. 1. *The authors find that wgn RNAi enhances hh>egr-induced apoptosis. They validate the results with two independent RNAi lines in Figure S1. However, Figure S1 is missing a control without wgn RNAi, and therefore, the results are difficult to assess.*

Fig S1A now contains this control.

2. 2. *Are the two independent wgn RNAi lines targeting different regions of the coding sequence?*

As the regions targeted by the 2 RNAi's are different, see below, we have included in the text:

“This observation was corroborated with an independent *RNAi-wgn* strain targeting a different region in the coding sequence (Fig. S1A and S1B). “

Bloomington BL55275 (dsRNA-HMCO3962)

VDRC GD9152 (dsRNA-GD3427)

2.7. In Figure 4, the authors show that *egr* expression induces ROS and performs anti-oxidant experiments. This part could be strengthened if they show that the ROS sensor signal disappears after *Sod::Cat* expression.

We had done this experiment and there is a definitely drop in Mitosox in discs in which the weak allele of *egr* is active. We have included this new image in Figure 4G and in the text.

2.8. How effective is *egr* RNAi? In Figure 5E, F, the authors knock down *egr* and obtain negative results. Based on this, the authors argue that *Wgn* localization occurs through an *egr*-independent mechanism. Drawing strong conclusions based on a negative result with *egr* RNAi is not a good practice since one cannot rule out residual *egr* activity that mediates the effect (of course, because there is cell death as well, death cells express *egr*). I suggest either finding ways to completely abolish *egr* function, or tone down the conclusion.

We have used 'after knocking down *egr*' instead of in the 'absence' or 'abolish' *egr*.

2.9. Figure 6 shows that *wgn* RNAi aggravates the reaper phenotype. What's missing is a control that expresses *wgn* RNAi but not reaper.

Control experiments using the *UAS-wgnRNAi* in the absence of *rpr* are now shown in figure S4C.

Reviewer #3

3.1. Minor Fig 6C-E would need a control disc without induced apoptosis (ie wildtype disc) stained for phospho-p38 as a baseline comparison. This is important to judge the significance of the remaining phospho-p38 in panel E where *wgn* is knocked down. The authors write "However, after knocking down *wgn*, phosphorylated p38 in the wing pouch surrounding the apoptotic cells was abolished (Fig. 6E)."

Depending on the amount of phospho-p38 in control discs, this may need to be rephrased to "strongly reduced" instead of "abolished".

A control disc stained with P-p38 has been added in Figure 6.

Revision Plan

We have changes 'abolished' by 'strongly reduced'.

3.2. *This sentence in the Intro needs fixing because TNF α doesn't transduce the signal from TNFR to Ask1 since it's upstream of TNFR:*

"TNF α can transduce the TRAF-mediated signal from TNFR to Ask1 to modulate its activity (Hoeflich et al., 1999; Nishitoh et al., 1998, p. 0; Obsil & Obsilova, 2017; Shiizaki et al., 2013)."

We have rephrased this sentence by:

"TNF α binds to TNFRs which in turn interact with TRAFs to transduce the signal to Ask1 to modulate its activity".

3.3a *In the results section, the authors start by ectopically overexpressing Eiger. Are there conditions where Eiger expression is induced in the wing? If yes, it would be helpful for the reader to mention that this system with the genetically GAL4-induced expression of Eiger aims to phenocopy one of these conditions.*

Eiger ectopic expression has been induced in the wing to generate apoptosis. This is a common technique in *Drosophila*, and the Reviewer3 is right that a sentence should be useful for the reader.

A sentence has been introduced at the beginning of the results section:

"Ectopic expression of *egr* in *Drosophila* imaginal discs results in JNK-dependent apoptosis (Brodsky et al., 2004; Igaki et al., 2002; Moreno, Yan, et al., 2002)."

3.3b *Fig 2C is not very self-explanatory: it is worth writing out what Hsa (*H. sapiens*), Bla and Sco stand for (there is plenty of space).*

We have re-designed figure 2 to make it more self-explanatory.

3.4. *This sentence is confusing:*

*"...Wgn localization were due to ROS or to the expression of *egr*, we used RNAi to knock down *egr* in the apoptotic cells and found that reduced Egr/TNF α had no effect on Wgn localization (Fig. 5E, 5F)."*

The authors may want to specify that Wgn is still accumulated even without Egr. ("No effect" is unclear).

This sentence has been modified as:

"Wgn localization were due to ROS or to the expression of *egr*, we used RNAi to knock down *egr* in the apoptotic cells and found that Wgn accumulation was not altered by the knocking down Egr/TNF α (Fig. 5E, 5F). "

3.5 *Comment. It discovers that Wengen is activated by ROS. In fact, since Wengen binds TNF with an affinity that is several orders of magnitude lower than Grindelwald, and since Wengen is not even located at the cell membrane, these data call into question whether Wengen is a TNF receptor, or a ROS receptor? Could the authors comment on this ? Could it be that the results obtained in the past showing that Wengen is activated by TNF were actually due to TNF*

inducing apoptosis, leading to production of ROS, leading to activation of Wengen?

We totally agree with Reviewer#3. We have added a final paragraph in the discussion section. “We speculate that the subcellular location of Wgn and Grnd may contribute to the different functions of both receptors. Grnd is more exposed at the apical side of the plasma membrane, which makes this receptor more accessible for ligand interactions (Palmerini et al., 2021). Wgn, embedded in cytoplasmic vesicles, is less accessible to the ligand and could be more restricted to being activated by local sources of signaling molecules, such as ROS. In contrast to initial reports (Kanda et al., 2002; Kauppila et al., 2003), los-of-function of wgn does not rescue Egr-induced apoptosis in the Drosophila eye (Andersen et al., 2015), which supports our observation in the wing that Wgn is not required for Egr-induced apoptosis. Instead, Egr-induced apoptosis generates ROS which target intracellular Wgn to foster a cell survival program of cells close to the apoptotic zone.”

Reviewer #4

4.1 b Are phospho-p38 levels increased in cells expressing Egr[weak]?

We have the results of these experiments. To respond to this point, a new figure has been added (Fig. S4) in which we show the P-p38 levels are increased (non-autonomously) in egr^w , as previously found for *reaper*. In addition, we show that egr^w + activation of p35 and egr^w + activation of Sod1::Cat results in strong reduction of P-p38. This indicates that P-p38 is stimulated by the ROS produced by apoptotic cells.

The text now:

“It is worth noting that cells egr^w induce phosphorylation of p38 in neighboring cells (Fig. S4A) and that, as previously found for *rpr* (REF), depends on ROS generated by egr^w apoptotic cells (Fig. S4B, C).”

4.2 In Figure 4C it appears that the Dcp-1 positive cells move apically rather than basally. Including nuclear staining would be very informative allowing assessment of tissue morphology. The authors focus on the pouch region of the wing imaginal disc, where phenotypes are strong and obvious. However, the hh-Gal4 driver also affects posterior cells in the hinge and notum, where the effects of Eiger[weak] overexpression seem weaker (e.g., minimal to no MitoSox signal in hinge and notum posterior cells). Could the authors explain this observation?

Point 1: Actually, cells move more basally, though some move more apical as well. Depending on the section cells the image could be confusing. To solve that, we show now a plane on these discs at apical and a plane basal. Both high magnifications. There one can see that there is more concentration of pyknotic nuclei basally. We have added this observation in a new supplementary figure (Fig. S3) and the corresponding text in page 5: “Apoptotic cells in egr^{weak} are characterized by pyknotic nuclei and are positive for Dcp1. These cells tend to concentrate in the basal side of the epithelium, although some are scattered apically (Fig. S3). Accumulation of Wgn was observed in healthy anterior cells adjacent to both apical and basal egr^{weak} cells (Fig. 4, Fig. S3A, B).”

Revision Plan

Point 2 Comment on MitoSOX in notum: At the stages of the imaginal discs used in this study, almost all notum cells are anterior compartment. The hh positive cells in notum much less abundant, therefore most of the staining was found in the posterior compartment of the wing pouch.

4.5 Figure 6 C-E. Does WgnRNAi potentiates and GrndRNAi suppress Rpr-induced apoptosis similarly to their effects when knocked down in Eiger[weak]OE cells?

The areas controlled by $sal^{E/Pv} > rpr$ (dotted lines) are full of pycnotic nuclei, which indicates concentration of apoptotic cells in all genotypes shown.

Thus, in the conditions generated here, apoptosis is not inhibited and grnd RNAi does not interfere with the activation of P-p38. In *wgn* knock down, phospho-p38 is strongly inhibited, indicating the importance of *wgn* in phosphorylation of p38.

To clarify this point, we have added in the text: "Note that *rpr*-induced apoptosis is not suppressed after knocking down *grnd* or *wgn*." Also in the figure legend we added: "White lines in the confocal images outline the $sal^{E/Pv}$ -LHG, LexO-*rpr* dark area full of pyknotic nuclei of apoptotic cells."

4.6 The activation of p38 following $salE/Pv > rpr$ -mediated ablation as shown by immunostaining is noteworthy. While loss Grnd knockdown leads to phospho-p38 signal enrichment around the rpr-expressing cells, WgnRNAi results in reduced phospho-p38 signal in the wing pouch but also beyond the nub-expression domain. Do $salE/Pv > rpr$ nub > WgnRNAi cells still generate ROS?

So far there is no evidence of Wengen as a ROS scavenger. We have evidence that ROS (using MitoSox probe) are produced in $egr^{weak} + Wgn$ RNAi cells. Thus, the inhibition of *wgn* expression does not block ROS production. A new figure shows this observation (Figure S4A).

4.7 Are ROS responsible for the long-range signaling and p38 activation, referring to authors' previous work Santabarbara-Ruiz et al., 2019, PLoS Genet 15(1):

e1007926. <https://doi.org/10.1371/journal.pgen.1007926>, Figure 5G?

ROS are responsible for p38 activation as shown in a new figure (Fig. S4). In this new figure *egrweak* is activated in hh, and p38 is most of cells in the posterior compartment, and also anterior. However, after blocking apoptosis or ROS production, this P-p38 is reduced.

4.8 Minor I propose rephrasing the description of "UAS-Egr[weak] transgene, a strain that produces a reduced Egr/TNF α activity". It could imply a loss of function strain rather than a transgene that causes mild/moderate Egr overexpression.

The sentence has been rephrases as suggested (End of the first paragraph in results section).

4.9 Minor. I recommend the authors to revise the charts for improved clarity in genotype

Revision Plan

representation. For example, in Figure 1I, the label "control-GFP" might be misleading. It would be beneficial to specify that "control" refers to Eiger[weak] alone with other manipulations being done simultaneously with Eiger[weak] overexpression.

All charts have been revised.

4.10 Minor. Additionally, considering that individuals with color blindness may struggle to differentiate between red and green colors, I strongly suggest using a color-blind-friendly palette, especially in Figure 4A, C, G, and Figure 4A, C, E."

All images have been revised for color blind code.

4.11 Minor. Providing detailed information regarding the reagents used in the study, such as Catalogue Numbers or RRIDs, is beneficial for enhancing reproducibility.

We have added the RRID and Cat #. If no ID was available, we added the reference or contact.

4.12 This reviewer points two limitations that we are now trying to solve:

Limitations:

Quality of the imaging – higher magnification images and quantification would enhance the study.

The summarizing model may contain excessive speculations that lack support from the data or references to the existing literature.

Quality of imaging. We have now an extra supplemental figure with higher magnifications. Extra higher magnifications will be included in the next version as well as quantification, as exposed for the Revision Plan points 4.3 and 4.4.

Model: We have re-written the paragraph on the model, introduced references and drop some speculations. We hope the current version will be more convincing for the reader.

4. Description of analyses that authors prefer not to carry out

Please include a point-by-point response explaining why some of the requested data or additional analyses might not be necessary or cannot be provided within the scope of a revision. This can be due to time or resource limitations or in case of disagreement about the necessity of such additional data given the scope of the study. Please leave empty if not applicable.

Reviewer 1

1.3. *Is the overexpression of Wengen sufficient to increase tissue regeneration?*

The suggestion of the reviewer is a key point in regeneration biology: how to accelerate regeneration?

We have demonstrated that Wengen is upstream the Ask1-p38 axis that drives regeneration.

The reviewer wonders if Wengen overexpression can result in increase in regeneration. In a previous work we have demonstrated that p38 activation is key for regeneration but its overexpression can be deleterious (Esteban-Collado et al., 2021). Only in discs that sensitized

Revision Plan

for low p38 (starvation, low Akt, Ask1S83A mutant), the overexpression rescues regeneration. Therefore, the levels of the Wgn-Ask1-p38 have to be very tightly controlled. An excess will be deleterious. We are aware of the importance of the question, but at this point we do not have the technology to finely control Wgn-Ask1-p38 levels to do this experiment.

1.6 Minor - It possible to test the induction of apoptosis in a wgn null mutant background to see if the phenotype is even stronger than the one observed with RNAi (the wgn RNAi is induced at the same time than egr or rpr overexpression).

Flies *wgn*^{KO} survive, but they gave us problems when carrying transgenes for our design of genetic ablation. Indeed, we tried to generate *wgn*^{KO} carrying *Gal4+tubGal80+eiger*^{weak} without success.

In addition, the reason we have used an inducible mutant is because it allows us to work in time and space without altering expression in other cell types beyond wing discs. Wgn is required in other organs during development like gut, trachea and axon growth, etc., and thus, we ensure the affected cells belong to the tissue analyzed.

Dear Dr. Serras,

Thank you for submitting your manuscript for consideration by the EMBO Journal. I have now read your manuscript, the reviewer comments and your response to them. Based on our editorial assessment and the referees' positive evaluations, I would like to invite you to submit a revised version of the manuscript along the lines indicated in your revision plan.

We generally allow three months as standard revision time. As a matter of policy, competing manuscripts published during this period will not negatively impact on our assessment of the conceptual advance presented by your study. However, please contact me as soon as possible upon publication of any related work to discuss the appropriate course of action. Should you foresee a problem in meeting this three-month deadline, please let us know in advance in order to arrange an extension.

When preparing your letter of response to the referees' comments, please bear in mind that this will form part of the Review Process File and will therefore be available online to the community. For more details on our Transparent Editorial Process, please visit our website: <https://www.embopress.org/page/journal/14602075/authorguide#transparentprocess>. Please also see the attached instructions for further guidelines on preparation of the revised manuscript.

Please feel free to contact me if you have any further questions regarding the revision. Thank you for the opportunity to consider your work for publication. I look forward to receiving your revised manuscript.

Yours sincerely,

Ieva Gailite

Please remember: Digital image enhancement is acceptable practice, as long as it accurately represents the original data and conforms to community standards. If a figure has been subjected to significant electronic manipulation, this must be noted in the

figure legend or in the 'Materials and Methods' section. The editors reserve the right to request original versions of figures and the original images that were used to assemble the figure.

We realize that it is difficult to revise to a specific deadline. In the interest of protecting the conceptual advance provided by the work, we recommend a revision within 3 months (5th May 2024). Please discuss the revision progress ahead of this time with the editor if you require more time to complete the revisions.

Rev_Com_number: RC-2023-02257

New_manu_number: EMBOJ-2024-116851

Corr_author: Serras

Title: ROS-mediated TNFR Wengen activation in response to apoptosis

Response to Reviewers Esteban Collado et al

Manuscript number: EMBOJ-2024-116851R (Number for Review Commons: #RC-2023-02257)

Corresponding author(s): Florenci SERRAS

General Statement

We are pleased to report that all four reviewers have provided positive feedback on the paper, along with some constructive suggestions to further enhance the manuscript and bolster our observations. We wholeheartedly acknowledge and embrace their recommendations. Consequently, we have diligently incorporated their feedback into the revised text and conducted additional experiments as outlined below. We wish to express our sincerest gratitude to the reviewers for their invaluable insights and contributions.

Reviewer #1

1.1 a- *The result in Fig1.H is somehow surprising. How does the overexpression of Egr induce caspase activation in the absence of its receptor Grnd?*

The results of Fig. 1H, in which *egr+grndRNAi+wgnRNAi* results in high apoptosis indicates that *wgn* downregulation compromises survival even in the absence of *grnd*. The reviewer correctly points that:

“How does the overexpression of Egr induce caspase activation in the absence of its receptor Grnd?”.

There is evidence that Eiger is involved in the regulation of the pro-apoptotic gene *head involution defective (hid)* in primordial germ cells (Maezawa 2009 Dev. Growth Differ., 51 (4) (2009), pp. 453-461) and in the elimination of damaged neurons during development (Shklover et al., 2015). Moreover, Eiger is necessary for HID stabilization and can regulate HID-induced apoptosis independently of JNK signaling (Shklover et al., 2015). Therefore, in our discs *egr* activation in the absence of *grnd* and *wgn* can still result in apoptosis because of the absence of *wgn*'s survival signal and, presumably, by activation of *hid*.

We have introduced this issue in the text as:

“To check for epistasis between *grnd* and *wgn*, we activated *hh> egrweak* and knocked down both TNFRs. We found high levels of cell death compared to *wgn* RNAi alone (Fig. 1H and 1I), which demonstrates that *wgn* down-regulation is dominant over *grnd*. This is surprising as it is generally assumed that Egr interacts with Grnd to induce apoptosis via JNK, which in turn activates the proapoptotic gene *hid* (Andersen et al., 2015; Diwanji & Bergmann, 2020; Fogarty et al., 2016; Igaki et al., 2002; Moreno, Yan, et al., 2002; Sanchez et al., 2019; Shlevkov & Morata, 2012). Interestingly, Egr is necessary for HID stabilization and can regulate HID-induced apoptosis independently of JNK (Shklover et al., 2015). Therefore, cells *egrweak* that downregulate *grnd* and *wgn* can still be eliminated because the lack of both *Wgn*-survival signal and the pro-apoptotic *Grnd*/JNK signal could result in an alternative pathway of apoptosis.”

Response to Reviewers Esteban Collado et al

1.1 b. Is it possible that the loss of function of Wengen on its own has a phenotype? If so, that would suggest that Wgn in addition to its role in regeneration might be implicated in pro-survival processes in homeostatic conditions?

This issue is very important to understand the differential role of Wgn and Grnd. First of all, Wengen knock out (*wgn^{KO}*; Andersen et al., 2015) is viable in homozygosis. Therefore, we have now crossed the flies to get the genotype *hh-Gal UAS RNAi wgn* of two independent lines and we have checked for apoptotic phenotype, as suggested by reviewer 1. We have found only very few apoptotic cells, which suggests that *wgn* is recruited under stress conditions rather than homeostasis. These results are now shown in the new Figure Extended View 1DE.

1.2- In Fig.6, it would be relevant to include wengen inactivation within the domain where rpr is expressed to show that Wengen is not required autonomously for regeneration (sal>rpr + wgn RNAi). What is the phenotype of the adult wing of sal-lexA>rpr + nub-gal4 >wgn RNAi animals.?

We have already added a new figure (Fig. Extended View 4C) containing this data. As shown, both *wgnRNAi* alone and *wgn RNAi + rpr* expressed both the Sal domain, do not show relevant anomalies and regenerate normally. Therefore, we conclude that *wgn* is not autonomously required for regeneration.

Instead, the adult wings *sal-lexA>rpr + nub-gal4 >wgn RNAi* result in a strong aberration, as regeneration is inhibited. This experiment has been also added in another figure (Fig. Extended View 4B).

1.3. Is the overexpression of Wengen sufficient to increase tissue regeneration?

The suggestion of the reviewer is a key point in regeneration biology: how to accelerate regeneration?

Thank you for your insightful suggestion regarding the potential impact of Wengen overexpression on regeneration. We appreciate your keen attention to our work. Our previous findings have highlighted the delicate balance required for optimal regeneration, particularly concerning the Wengen-Ask1-p38 axis. In our previous study (Esteban-Collado et al., 2021), we elucidated that while p38 activation is crucial for regeneration, its overexpression can have deleterious effects, except in specific contexts where p38 sensitization occurs.

We agree that tightly controlling the levels of Wengen-Ask1-p38 is paramount for understanding the intricacies of regeneration. However, at this stage, we acknowledge the need for further refinement of our binary expression systems to precisely regulate these levels. Improving the dual expression systems will allow us to conduct the suggested experiment with the necessary precision and control. We are fully committed to addressing this important question in our future studies.

Once again, we sincerely appreciate your insightful input and your understanding of the complexities involved in our research. Your feedback has provided valuable guidance for our ongoing investigations.

1.4 Minor- In fig. 11, it is surprising that knockdown of neither Grnd nor dTRAF2 significantly affects Egr-induced apoptosis

Response to Reviewers Esteban Collado et al

After applying a One-Way ANOVA test to compare all the groups against all the groups in fig. 1B no significant differences were detected between control and RNAi grnd or RNAi dTRAF2 ($p > 0,05$). But if we apply a Student's T test, which is less restrictive, we obtain, indeed, significant differences:

Control vs. RNAigrnd $p = 9,48 \times 10^{-7}$

Control vs. RNAi dTRAF2 $p = 2,47 \times 10^{-7}$

We have now added in the text:

“Note that when egrweak cells downregulated dTRAF2 or grnd the cell death area ratio is slightly lower than egrweak alone (Fig. 1I), confirming that dTRAF2 and Grnd contribute to apoptosis in egrweak cells.”

1.5 Minor The ability of the wing disc to regenerate has been associated with the induction of a developmental delay mediated by Dilp8. Are the authors observing this developmental delay in the case of sal-lexA>rpr + Ap-gal4 >wgn RNAi or sal-lexA>rpr + Ci-Gal4>wgn RNAi

The developmental delay due to Dilp8 has been observed by many laboratories, indeed. The question of the reviewer is relevant because if there is no delay in pupariation, regeneration could be compromised not because regeneration has been affected but because after pupariation regeneration is impeded.

However, delay in pupariation has been found in our experiments. Usually for 11hrs of heat shock (to induce apoptosis) we found 1-2 days of delay.

We have added the following text:

“The ability of the wing disc to regenerate after genetic ablation has been associated with the induction of a developmental delay (Colombani et al., 2012; Garelli et al., 2012; Jaszczak et al., 2015; Katsuyama et al., 2015; Smith-Bolton et al., 2009). All genotypes analyzed in figure 6 showed a similar developmental delay of 1-2 days (at 17°C) after genetic ablation in comparison to the animals of the same genotype in which no genetic ablation was induced, i.e. developed continuously at 17°C (Fig. EV6A). After the adults emerged, the wings were dissected, and regeneration was analyzed.”

1.6 Minor - It possible to test the induction of apoptosis in a wgn null mutant background to see if the phenotype is even stronger than the one observed with RNAi (the wgn RNAi is induced at the same time than egr or rpr overexpression).

Flies wgn^{KO} survive, but they gave us problems when carrying transgenes for our design of genetic ablation. Indeed, we tried to generate wgn^{KO} carrying $Gal4+tubGal80+eiger^{weak}$ without success.

In addition, the reason we have used an inducible mutant is because it allows us to work in time and space without altering expression in other cell types beyond wing discs. Wgn is required in other organs during development like gut, trachea and axon growth, etc., and thus, we ensure the affected cells belong to the tissue analyzed.

1.7 Minor - The investigation of the evolutionary origin of TNFR in drosophila included in Fig.2 is cutting a bit the flow of the results.

Response to Reviewers Esteban Collado et al

The evolutionary origin starts now with a sentence that can smoothen the flow and few changes in that paragraph have been made:

“Opposing roles between proteins of the TNFR superfamily suggests that they have an ancient origin and have followed divergent evolutionary paths. To track the differences observed between *grnd* and *wgn*, we decided to investigate the evolutionary origin of these two *Drosophila* genes.”

1.8 Minor *The authors could explain in more details the double transactivation system for non-fly specialists.*

The entire section has been re-written in Material and Methods.

1.9 Minor - *It can be interesting to include and/or discuss these few references:*

PLoS Genet. 2019 Aug; 15(8): e1008133.

PLoS Genet. 2022 Dec 5;18(12):e1010533.

FEBS Lett. 2023 Oct;597(19):2416-2432.

Curr Biol. 2016 Mar 7;26(5):575-84.

Nat Commun. 2020 Jul 20;11(1):3631.

All these references have been introduced in the text.

Reviewer #2

2. 1. *The authors find that wgn RNAi enhances hh>egr-induced apoptosis. They validate the results with two independent RNAi lines in Figure S1. However, Figure S1 is missing a control without wgn RNAi, and therefore, the results are difficult to assess.*

This figure, Figure Extended View 1A, now contains this control.

2. 2. *Are the two independent wgn RNAi lines targeting different regions of the coding sequence?*

As the regions targeted by the 2 RNAi's are different, see below, we have included in the text: “This observation was corroborated with an independent *RNAi-wgn* strain targeting a different region in the coding sequence (Fig. Extended View 1D and E). “

Bloomington BL55275 (dsRNA-HMCO3962) (TRIP)

VDRC GD9152 (dsRNA-GD3427)

Response to Reviewers Esteban Collado et al

2.3. Aside from *wgn*, other RNAi experiments are not validated through independent RNAi lines. I suggest expanding the Supplemental Figures to reproduce a few key findings with independent RNAi lines.

We have recently acquired a set of independent RNAi lines from both the Bloomington Stock Center and the Vienna Drosophila Resource Center. Initially, we did not pursue this avenue as we were primarily focused on investigating the roles of *Wgn* and *Grnd*. However, we wholeheartedly agree with the suggestion made by Reviewer 2, prompting us to include the results of these pivotal experiments with independent lines in Figure 1.

Our thorough analysis has revealed that these newly acquired lines demonstrate behavior consistent with those presented in the previous version of our study. The additional RNAi strains utilized for all genes under investigation serve to validate the significance of the *Wgn*/dTRAF1/Ask1 axis in survival and the *Grnd*/dTRAF2/Tak1 axis in apoptosis.

We have seamlessly integrated these new findings into the graphic representation of Figure 11. Moreover, individual images of these newly incorporated RNAi lines are now available in Figure Extended View 1 (F, G, H, I, J). We believe that these additions enrich the comprehensiveness of our study and sincerely appreciate the opportunity to enhance its clarity and depth.

2.4. In Figure 1E, the authors show that *wgn* RNAi enhances cell death caused by *hh>egr*. What is missing here is a *wgn* RNAi control without *hh>egr*. Is there any cell death caused by the loss of *wgn* alone (without *hh>egr*)?

Wgn alone has no phenotype, or few scattered apoptotic cells. Therefore, this observation reinforces that *wgn* is recruited for a stress- dependent function.

As responded to Reviewer 1, this issue is very important to understand the differential role of *Wgn* and *Grnd*. First of all, *Wgn* knock out (*wgn*^{KO}; Andersen et al., 2015) is viable in homozygosis. Therefore, we have now crossed the flies to get the genotype *hh-Gal UAS RNAi wgn* of two independent lines and we have checked for apoptotic phenotype, as suggested. We have found that very few apoptotic cells. Instead, this experiment suggests that *wgn* is recruited under stress conditions rather than homeostasis. These results are now shown in the new Figure Extended View 1DE.

2.5. If *wgn* RNAi causes some degree of cell death, is the observed effect with *hh>egr* a significant genetic interaction, or merely additive?

As described in the previous comment the number of apoptotic cells is very low, suggesting a role of *Wgn* in stress- or damage-dependent context.

Response to Reviewers Esteban Collado et al

2. 6. *Is the wgn-p38 pathway sufficient to block egr induced cell death? The authors could test this by having hh>egr in the licT1.1 background. The authors have a more complex experiment in Figure 3, where licT1.1 is introduced into the hh>egr, wgnRNAi background. However, testing the effect of licT1.1 without wgn would establish a more direct relationship between egr and wgn-p38.*

We agree with the reviewer that examining *lic* expression in an *egr* activated background could offer valuable insights. However, our investigations have revealed that co-expression of *lic* and *egr* leads to increased cell death. This outcome is likely due to an excess of phospho-p38, as it is generated both in response to apoptosis and as a result of *lic* expression. Consequently, an overabundance of p38 can prove detrimental to the cells, ultimately culminating in apoptosis. This observation aligns with our previous findings, where we noted the sensitivity of p38 activity – both low and high levels can be lethal for the cell.

We have now incorporated these important observations into Figure Extended View 3. Furthermore, our experiments have shown that downregulation of p38 also results in cell death. This underscores the critical importance of finely regulating p38 levels.

2.7. *In Figure 4, the authors show that egr expression induces ROS and performs anti-oxidant experiments. This part could be strengthened if they show that the ROS sensor signal disappears after Sod::Cat expression.*

We have done this experiment and there is a definitively drop in Mitosox in discs in which the weak allele of *egr* and *Sod::Cat* are co-expressed. We have included this new image in Figure 4G and in the text.

2.8. *How effective is egr RNAi? In Figure 5E, F, the authors knock down egr and obtain negative results. Based on this, the authors argue that Wgn localization occurs through an egr-independent mechanism. Drawing strong conclusions based on a negative result with egr RNAi is not a good practice since one cannot rule out residual egr activity that mediates the effect (of course, because there is cell death as well, death cells express egr). I suggest either finding ways to completely abolish egr function, or tone down the conclusion.*

We agree also in this issue. We have changed the text and now have used ‘after knocking down eiger’ instead of ‘in the absence’ or ‘abolish’ eiger.

2. 9. *Figure 6 shows that wgn RNAi aggravates the reaper phenotype. What's missing is a control that expresses wgn RNAi but not reaper.*

Control experiments using the *UAS-wgnRNAi* in the absence of *rpr* are now shown in figure Extended View 1A.

Reviewer #3

3.1. *Minor Fig 6C-E would need a control disc without induced apoptosis (ie wildtype disc) stained for phospho-p38 as a baseline comparison. This is important to judge the significance of the remaining phospho-p38 in panel E where wgn is knocked down. The authors write " However, after knocking down wgn, phosphorylated p38 in the wing pouch*

Response to Reviewers Esteban Collado et al

surrounding the apoptotic cells was abolished (Fig. 6E)."

Depending on the amount of phospho-p38 in control discs, this may need to be rephrased to "strongly reduced" instead of "abolished".

A control disc stained with P-p38 has been added in Figure 6C.

The Fig6E is now Fig 6F: We have changed 'abolished' by 'strongly reduced'.

3.2. This sentence in the Intro needs fixing because TNF α doesn't transduce the signal from TNFR to Ask1 since it's upstream of TNFR:

"TNF α can transduce the TRAF-mediated signal from TNFR to Ask1 to modulate its activity (Hoeflich et al., 1999; Nishitoh et al., 1998, p. 0; Obsil & Obsilova, 2017; Shiizaki et al., 2013)."

We have rephrased this sentence by:

"TNF α binds to TNFRs which in turn interact with TRAFs to transduce the signal to Ask1 to modulate its activity".

3.3a In the results section, the authors start by ectopically overexpressing Eiger. Are there conditions where Eiger expression is induced in the wing? If yes, it would be helpful for the reader to mention that this system with the genetically GAL4-induced expression of Eiger aims to phenocopy one of these conditions.

Eiger ectopic expression has been induced in the wing to generate apoptosis. This is a common technique in *Drosophila*, and the Reviewer3 is right that a sentence should be useful for the reader.

A sentence has been introduced at the beginning of the results section:

"Ectopic expression of *egr* in *Drosophila* imaginal discs results in JNK-dependent apoptosis (Brodsky et al., 2004; Igaki et al., 2002; Moreno, Yan, et al., 2002)."

*3.3b Fig 2C is not very self-explanatory: it is worth writing out what Hsa (*H. sapiens*), Bla and Sco stand for (there is plenty of space).*

We have re-designed figure 2 to make it more self-explanatory and used the common names and in the results section:

"...clustering together with some of the CRD sequences from hemichordates (the acorn worm, *Saccoglossus kowalevskii*), non-vertebrate chordates (the European amphioxus, *Branchiostoma lanceolatum*) and vertebrates (*Homo sapiens*).

3.4. This sentence is confusing:

*"...Wgn localization were due to ROS or to the expression of *egr*, we used RNAi to knock down *egr* in the apoptotic cells and found that reduced Egr/TNF α had no effect on Wgn localization (Fig. 5E, 5F)."*

The authors may want to specify that Wgn is still accumulated even without Egr. ("No effect" is unclear).

This sentence has been modified as:

"To test if the effects on Wgn localization were due to ROS or to the expression of *egr*, we used RNAi to knock down *egr* in the apoptotic cells and found that Wgn accumulation was not altered by the knocking down Egr/TNF α (Fig. 5E, 5F). "

Response to Reviewers Esteban Collado et al

3.5 Comment. It discovers that Wengen is activated by ROS. In fact, since Wengen binds TNF with an affinity that is several orders of magnitude lower than Grindelwald, and since Wengen is not even located at the cell membrane, these data call into question whether Wengen is a TNF receptor, or a ROS receptor? Could the authors comment on this ? Could it be that the results obtained in the past showing that Wengen is activated by TNF were actually due to TNF inducing apoptosis, leading to production of ROS, leading to activation of Wengen?

We totally agree with Reviewer#3. We have added this issue in the discussion section: “We speculate that the subcellular location of Wgn and Grnd may contribute to the different functions of both receptors. Grnd is more exposed at the apical side of the plasma membrane, which makes this receptor more accessible for ligand interactions (Palmerini et al., 2021) to transduce a JNK-dependent apoptotic signal (Moreno, Yan, et al., 2002; Shlevkov & Morata, 2012; Tobiume et al., 2001). Wgn, embedded in cytoplasmic vesicles, is less accessible to the ligand and could be more restricted to being activated by local sources of signaling molecules, such as ROS. In contrast to initial reports (Kanda et al., 2002; Kauppila et al., 2003), loss-of-function of wgn does not rescue Egr-induced apoptosis in the Drosophila eye (Andersen et al., 2015), which supports our observation in the wing that Wgn is not required for Egr-induced apoptosis. Instead, Egr-induced apoptosis generates ROS which target intracellular Wgn to foster a cell survival program of cells adjacent to the apoptotic zone.”

Reviewer #4

4.1a The conclusion regarding the protective role of p38 in response to Egr[weak] should be supported by a p38 knockdown experiment.

To demonstrate the protective role of p38 we have included now a figure (Figure Extended View 3A) showing high incidence of apoptosis in hh>egw RNAi p38. This observation supports the need of p38 for survival.

4.1 b Are phospho-p38 levels increased in cells expressing Egr[weak]?

To respond to this point, a new figure has been added (Fig. EV4D) in which we show the P-p38 levels are increased (non-autonomously) in *egr^w*, as previously described for *reaper*. In addition, we show that *egr^w* + activation of p35 and *egr^w* + activation of Sod1::Cat results in strong reduction of P-p38. This indicates that P-p38 is stimulated by the ROS produced by apoptotic cells.

The text now:

“It is worth noting that cells *egr^w* induce phosphorylation of p38 in neighboring cells (Fig. EV4D), and as occurs in *rpr*-expressing cells (Santabárbara-Ruiz et al., 2015). ROS depletion and p35-expression reduce phospho-p38 (Fig. EV4E,F). This suggests that activation of p38 depends on ROS generated by *egr^{weak}* apoptotic cells.”

4.2 In Figure 4C it appears that the Dcp-1 positive cells move apically rather than basally. Including nuclear staining would be very informative allowing assessment of tissue morphology.

Response to Reviewers Esteban Collado et al

The authors focus on the pouch region of the wing imaginal disc, where phenotypes are strong and obvious. However, the hh-Gal4 driver also affects posterior cells in the hinge and notum, where the effects of Eiger[weak] overexpression seem weaker (e.g., minimal to no MitoSox signal in hinge and notum posterior cells). Could the authors explain this observation?

Point 1: Actually, *egr* cells move more basally, though some move more apical as well. Depending on the section the image could be confusing. To solve that, we show now a plane on these discs at apical and a plane basal. Both high magnifications. There, one can see that there is more concentration of pyknotic nuclei basally. We have added this observation in a new supplementary figure (Fig. EV4 AB) and the corresponding text in page 5: "Apoptotic cells in *egr^{weak}* are characterized by pyknotic nuclei and are positive for Dcp1. These cells can be found along the apical-basal axis, although eventually tend to concentrate in the basal side of the epithelium (Fig. EV4A,B). Accumulation of Wgn was observed in healthy anterior cells adjacent to both apical and basal *egr^{weak}* cells (Fig. 4, and Fig. EV4A,B). By contrast, Grnd was maintained on the apical membrane of the anterior compartment and was found internalized in the *egr^{weak}* posterior compartment (Fig. 4D), which agrees with previous observations that Grnd is translocated from the membrane after binding to Egr/TNF α (Andersen et al., 2015) "

Point 2 Comment on MitoSOX in notum: At the stages of the imaginal discs used in this study, most notum cells are anterior compartment. The hh positive cells in notum much less abundant, therefore most of the staining was found in the posterior compartment of the wing pouch.

Major comments:

4.3 In Figure 5, the cells expressing Rpr appeared to be pulled/extruded basally as expected. It would be beneficial to quantify Wgn and Grnd signals along cross-sections and provide higher magnification images of domain boundaries to illustrate differences in TNFR localization and levels.

High magnification images are now in Figure Extended View 5. In these figures the boundary between *rpr*-induced cell death and Wengen tissue can be easily observed. We have quantified Wgn and Grnd localization and showed the data in Figure 5G and H. Data has been collected in ROI's in the A-compartment and P-compartment for each disc of all three conditions. Only the intensity of Wgn in *egr+Sod1:Cat* drops. This result strengthens the ROS-dependence of Wgn.

*The micrographs for Grnd Figure 5B,D, F capture substantial signal from the peripodial epithelium where the *salE/Pv*> driver is likely not active?*

Yes, Grnd is also detected in the peripodial membrane, even in regions far from the driver. Note that the staining of Grnd is localized on the apical membrane, and does not vary in these genetics conditions. Wgn shows a marked decrease upon expression of ROS scavengers (see Figure 5G and H).

4.4 The non-autonomous induction of Wgn seems stronger when facing dying Rpr

Response to Reviewers Esteban Collado et al

overexpressing cells simultaneously depleted of Eiger compared to Rpr OE alone. Should this be a reproducible, could the authors discuss potential reason for this observation?

It is difficult to respond this question without quantification. Therefore, we can now say that from the quantification presented in the the previous point, we conclude that there are no significant differences (Figure 5G and H).

4.5 Figure 6 C-E. Does WgnRNAi potentiates and GrndRNAi suppress Rpr-induced apoptosis similarly to their effects when knocked down in Eiger[weak]OE cells?

The areas controlled by $sal^{E/Pv} > rpr$ (dotted lines) are full of pycnotic nuclei, which indicates concentration of apoptotic cells in all genotypes shown.

Thus, in the conditions generated here, apoptosis is not inhibited and *grnd* RNAi does not interfere with the activation of P-p38. In *wgn* knock down, phospho-p38 is strongly inhibited, indicating the importance of *wgn* in phosphorylation of p38.

To clarify this point, we have added in the text:

“The accumulation of pyknotic nuclei after *rpr* ectopic expression, indicates that apoptosis is not suppressed after knocking down *grnd* or *wgn*.”

Also in the figure legend we added:

“White lines in the confocal images outline the $sal^{E/Pv}$ -LHG, LexO-*rpr* dark area full of pyknotic nuclei of apoptotic cells.”

4.6 The activation of p38 following salE/Pv>rpr-mediated ablation as shown by immunostaining is noteworthy. While loss Grnd knockdown leads to phospho-p38 signal enrichment around the rpr-expressing cells, WgnRNAi results in reduced phospho-p38 signal in the wing pouch but also beyond the nub-expression domain. Do salE/Pv>rpr nub>WgnRNAi cells still generate ROS?

So far there is no evidence of Wengen as a ROS scavenger. We have checked this issue and found evidence that ROS (using MitoSox probe) are produced in $egr^{weak} + Wgn$ RNAi cells. Thus, the inhibition of *wgn* expression does not block ROS production. A new figure shows this observation (Figure EV4C).

4.7 Are ROS responsible for the long-range signaling and p38 activation, referring to authors' previous work Santabarbara-Ruiz et al., 2019, PLoS Genet 15(1):

e1007926. <https://doi.org/10.1371/journal.pgen.1007926>, Figure 5G?

We think that very likely they are. ROS are responsible for p38 activation as shown in a new figure. In this new figure we show that when egr^{weak} is activated under *hh-Gal4*, and phospho-p38 localizes in the posterior compartment and also anterior, demonstrating that the response is not autonomous. In other words, cells surrounding a region in apoptosis will phosphorylate p38. Remarkably, after blocking ROS production (Sod1::Cat) or apoptosis (p35) this P-p38 is reduced. This demonstrates that the (non-autonomous) activation of p38 depends on ROS produced by the apoptotic cells.

This issue is now in the text and in Figure Extended View 4 D,E,F.

4.8 Minor I propose rephrasing the description of "UAS-Egr[weak] transgene, a strain that

Response to Reviewers Esteban Collado et al

produces a reduced Egr/TNF α activity". It could imply a loss of function strain rather than a transgene that causes mild/moderate Egr overexpression.

The sentence has been rephrased as suggested (End of the first paragraph in results section):

"We used the *hedgehog-Gal4* strain (hereafter *hh*>) to activate the transcription of *egr* in the posterior compartment using the *UAS-egr^{weak}* strain (hereafter *egr^{weak}*), a transgene that causes mild/moderate *egr* overexpression (Moreno, Basler, et al., 2002). "

4.9 Minor. I recommend the authors to revise the charts for improved clarity in genotype representation. For example, in Figure 1I, the label "control-GFP" might be misleading. It would be beneficial to specify that "control" refers to Eiger[weak] alone with other manipulations being done simultaneously with Eiger[weak] overexpression.

All charts have been revised.

4.10 Minor. Additionally, considering that individuals with color blindness may struggle to differentiate between red and green colors, I strongly suggest using a color-blind-friendly palette, especially in Figure 4A, C, G, and Figure 4A, C, E."

All images have been revised for color blind code.

4.11 Minor. Providing detailed information regarding the reagents used in the study, such as Catalogue Numbers or RRIDs, is beneficial for enhancing reproducibility.

We have added the RRID and Cat #. If no ID was available, we added the reference or contact.

4.12 This reviewer points two limitations that we are now trying to solve:

Limitations:

Quality of the imaging – higher magnification images and quantification would enhance the study.

The summarizing model may contain excessive speculations that lack support from the data or references to the existing literature.

Quality of imaging. We have now an extra figures with higher magnifications (Figures Extended View 4 and 5).

Quantification has been added. Notably, in Figure 5, the ROS-dependance of Wgn has been now quantified after down-regulating *egr* or promoting ROS scavenging (Sod:Cat).

About the Model in Figure 7: We have re-written the paragraph on the model, introduced references and drop some speculations.

We hope the current version will be more convincing for the reviewer and for readers.

Dear Florenci,

Thank you for submitting the revised version of your manuscript to The EMBO Journal. Your study has now been seen by all original referees, who find that their main concerns have been addressed and now broadly recommend acceptance of the manuscript.

In addition to the minor referee points, there now remain only a few editorial aspects that need addressing before I can extend formal acceptance of the manuscript:

1. In the Author Checklist file, please fill in the "Sample definition and in-laboratory replication" section (rows 87-88).
2. Please update the nomenclature to that of Figure EV1-EV6 (i.e., in the figure legend section) and Table EV1, also in the manuscript text. There is a reference to figure S6B on page 11 - please correct.
3. Please replace Figures EV1 and EV3 with the corrected version.
4. The wing image for regenerated wing has been reused in figure panels Fig 6B, EV6B-C. Please add a note to the figure legends that the image is reused and serves an illustrative purpose.
5. Please remove the legend for Table EV1 from the manuscript text file and add it to the table file.
6. Please move Methods section after Discussion and Figure legends after References.
7. We require a Data Availability Section at the end of Materials and Methods. As far as I can see, no data deposition in external databases is needed for this paper. If I am correct, then please state in this section: This study includes no data deposited in external repositories. Further information can be found at <https://www.embopress.org/page/journal/14602075/authorguide#dataavailability>
8. Please update references according to The EMBO Journal style - where there are more than 10 authors on a paper, the first 10 should be listed, followed by 'et al.' Please also remove DOIs for published papers. Please find further information here: <https://www.embopress.org/page/journal/14602075/authorguide#referencesformat>
9. Our data editors have flagged the following issues in figure legends that need correcting:
 - Please indicate the statistical test used for data analysis in the legends of figures 1i; 5g; 6b; EV 6b.
 - Please define the box plots in terms of minima, maxima, centre, bounds of box and whiskers, and percentile in the legends of figures 5g-h; 6b; EV 6a-c.
 - Please note that information about the number and nature of replicates is missing in the legends of figures 5g-h.
10. Papers published in The EMBO Journal are accompanied online by a 'Synopsis' to enhance discoverability of the manuscript. Please submit a short (1-2 sentences) summary of the findings and their significance in addition to the already provided bullet points highlighting the key results. Please also send us a synopsis image that is 550x300-600 pixels large (width x height, jpeg or png format). You can either show a model or key data in the synopsis image. Please note that the image size is rather small and that text needs to be readable at the final size.

With best wishes,

leva

leva Gailite, PhD
Senior Scientific Editor
The EMBO Journal
Meyerhofstrasse 1
D-69117 Heidelberg
Tel: +4962218891309
i.gailite@embojournal.org

Revision to The EMBO Journal should be submitted online within 90 days, unless an extension has been requested and approved by the editor.

Referee #1:

Comments on the authors answers in Review Commons and newly incorporated experiments:

The authors have addressed most of my comments and the comments of the 3 others reviewers. I think that the paper is fulfilling the requirements for publication.

Minor point:

I am still a bit puzzled by the fact that the over expression of Egr induces caspase activation in the absence of its receptor Grnd and maybe not fully convinced by the argument of the authors.

Can the authors exclude the possibility that the production of Egr is faster than the silencing of Grnd? Since both UAS are induced at the same time, the full efficiency of the Grnd RNAi can take more time than the OE of Egr and that is why some apoptosis can still be detected? Moreover, the RNAi is probably not a full silencing as compared to a KO.

Referee #2:

This manuscript by Esteban-Collado and colleagues uncovers the two opposing roles of the *Drosophila* TNF receptors, Wengen (Wng) and Grnd. The authors demonstrate that Grnd promotes Eiger-induced cell death, whereas Wengen promotes cell survival and regeneration in *Drosophila* wing discs. They further show that Wgn exerts its pro-survival role by signaling through p38. Moreover, they show that Wng is regulated by ROS generated in response to caspase activation. Such ROS-regulated pro-survival role of Wng provides a framework to understand how tissues with massive apoptosis undergo regeneration.

Overall, the manuscript fills major gaps in our understanding of TNFR signaling in *Drosophila*. The authors have satisfactorily addressed all points that I had raised. I only have a very minor point for the authors' consideration:

In page 4: The authors write that double knockdown of grnd and wgn show enhanced apoptosis. They conclude that cells still die "due to both the lack of the Wgn survival signal and the ability of the pro-apoptotic Grnd/JNK signal to trigger an alternative apoptosis pathway."

There is yet another possibility that is not considered here. The experiments are based on RNAi knockdown, and one cannot rule out the presence of residual Grnd/JNK signal, which may be enough to cause cell death when there is a stronger pressure to trigger apoptosis. I suggest not making strong conclusions about "an alternative apoptosis pathway" based on negative data with RNAi experiments.

Otherwise, this is an excellent manuscript.

Referee #3:

This is an interesting and convincing work that demonstrates a distinct role for the two *Drosophila* TNFRs, Grindelwald (Grnd) and Wengen (Wgn) in the tissue regeneration using the model of the developing *Drosophila* wing imaginal discs. Using functional genetics, the authors provide genetic evidence for the requirement of the Wengen-dTraf1-Ask1 module in promoting cell survival and tissue regeneration in response to spatially and temporally restricted apoptosis-induced tissue ablation. They implicate apoptosis-induced ROS, but not Eiger upregulation, in a non-autonomous enrichment of Wgn and p38 activation, both required for tissue regeneration. In contrast, the Grnd-dTraf2-Tak1 axis promotes cell apoptosis in an Egr-dependent manner. The study significantly underscores the emerging evidence for functional divergence and ligand in/dependence of Grnd and Wgn observed in several recent studies (Palmerini et al., 2021, Laudhaief et al., 2023, Letizia et al., 2023, Ruan et al., 2013, Andersen et al., 2023), which the authors further support by phylogenetic analysis.

Authors have strengthened their conclusions by performing additional experiments and quantification as requested by the reviewers and have improved the image presentation throughout the manuscript.

My main concern is the flow of the results section and the logical order of the arguments and conclusions. The authors have done an excellent job of performing additional experiments, but their attempt to include all the data and the description has led to repetition in the text. The text is currently not the easiest to read. The significance of individual data is lost in repeated arguments and conclusions, particularly within the Results section: "Wgn is required for survival by protecting cells from apoptosis" and "The protective role of Wgn is mediated by the activation of p38 MAP kinase" with licWT results at the end of the chapter.

Moreover, I would strongly recommend moving the phylogenetic analysis part to the end of the results rather than letting it

interrupt the flow of genetic functional experiments.

Minor comments

Authors may want to consider placing the description of the image channels in the upper right corner of their micrographs, as in Fig. EV5A-C. Text in the lower right corner often obstructs the view of the cells of interest, e.g. in Figs. 1, 3, EV1...

Authors should consider using MAPKKK and MAPKK instead of MAP3 and MAP2 kinase, respectively, and provide the full name of the abbreviation the first time it is used in the text.

"The expression of *egr*[weak] resulted in low levels of apoptotic cells in the posterior compartment of the disc (Figs. 1A,I, and EV1A). However, *egr*[weak] expression resulted in a strong abolition of apoptosis when *grnd* was downregulated (Figs. 1B,I, and EV1F), which coincides with previous findings that *Grnd* promotes JNK-dependent apoptosis (Andersen et al., 2015; Palmerini et al., 2021)."

The authors might want to omit the adjective "strong" as they describe the level of apoptosis as low to begin with.

"Therefore, the massive *wgn*-related apoptosis occurs only in stressed conditions generated by *egr* expression, suggesting that *wgn* is recruited for a stress-dependent environment."

"Context" might be more suitable word instead of "environment".

"Interestingly, *Egr* is necessary for the stabilization of the pro-apoptotic gene *hid* and can regulate *HID*-induced apoptosis independently of JNK (Shklover et al., 2015). Therefore, *egr*weak cells in which *grnd* and *wgn* are downregulated can still be eliminated, due to both the lack of the *Wgn* survival signal and the ability of the pro-apoptotic *Grnd*/JNK signal to trigger an alternative apoptosis pathway."

Seems very speculative for the Results section. What alternative pathway do they have in mind?

"Together, these observations suggest that the *Wgn* response to apoptotic ROS production occurs independently of a reduction in *Egr*/TNF α ."

Based on the results presented, the conclusion should read "Together, these observations suggest that the *Wgn* response to apoptotic ROS production occurs independently of *Egr*/TNF α " since the authors see *Egr* upregulation in *rprOE* cells and reducing *Egr* does not change *Wgn* accumulation.

"This function is primarily involves exacerbating p38 activity, likely via dTRAF1/Ask1, and is tightly linked to the oxidative stress induced following cell damage."

"Wengen" was first discovered as a TNFR that is able to bind to TRAF, trigger JNK-dependent apoptosis, and transduce signals from *Egr*/TNF α (Geuking et al., 2005; Kanda et al., 2002; Kauppila et al., 2003)."

Please correct these two sentences.

"In figure S6B we used a two-tailed Student's t-test to make the statistical comparisons using GraphPad (**** $p < 0.0001$)."
Should be Figure EV6B.

Rev_Com_number: RC-2023-02257

New_manu_number: EMBOJ-2024-116851R

Corr_author: Serras

Title: ROS-mediated TNFR Wengen activation in response to apoptosis

The authors addressed the minor editorial issues.

Dear Florenci,

Thank you for addressing the final minor points. I sincerely apologise for the slow process from our side due to the high number of submissions that we experience at the moment. I am now pleased to inform you that your manuscript has been accepted for publication - congratulations on a nice study!

Before we forward your manuscript to our publishers, I would like to propose some minor edits in the manuscript title, abstract and synopsis (please see below and the attached manuscript text file). I have also written a short blurb that will accompany the title of your manuscript in our online system. Please let me know if any corrections or adjustments are needed:

Title:

Reactive oxygen species activate the Drosophila TNF receptor Wengen for damage-induced regeneration

Blurb:

The two Drosophila TNFRs perform opposing functions in response to apoptosis and differ in their dependence on TNF α for activation.

Synopsis:

TNF receptors (TNFR) have been associated with pleiotropic pro-inflammatory functions ranging from apoptosis to survival. This study shows that the two Drosophila TNFRs, Wengen (Wgn) and Grindelwald (Gwd), are functionally divergent, playing opposing roles in response to apoptosis.

- Wgn is required for cell survival independently of the TNF α /Eiger, whereas Grnd promotes TNF α /Eiger-induced apoptosis.
- Wgn exerts its pro-survival role by signaling through the MAPK p38 during regeneration.
- Wgn, but not Grnd, is regulated by ROS generated in response to damage in a non-autonomous manner.
- The specialization of grnd and wgn for specific biological functions is supported by phylogenetic studies on their evolutionary origin

If you have any questions, please do not hesitate to contact the Editorial Office. Thank you for this contribution to The EMBO Journal and congratulations on a great paper!

With best wishes,

leva

leva Gailite, PhD
Senior Scientific Editor
The EMBO Journal
Meyerohofstrasse 1
D-69117 Heidelberg
Tel: +4962218891309
i.gailite@embojournal.org

Rev_Com_number: RC-2023-02257

New_manu_number: EMBOJ-2024-116851R1

Corr_author: Serras

Title: ROS-mediated TNFR Wengen activation in response to apoptosis